# SwinTrack: A Simple and Strong Baseline for Transformer Tracking

**Liting Lin**[1,2*]   **Heng Fan**[3*]   **Zhipeng Zhang**[4]   **Yong Xu**[1,2]   **Haibin Ling**[5]

[1]School of Computer Science & Engineering, South China Univ. of Tech., Guangzhou, China
[2]Peng Cheng Laboratory, Shenzhen, China
[3]Department of Computer Science and Engineering, University of North Texas, Denton, USA
[4]DiDi Chuxing, Beijing, China
[5]Department of Computer Science, Stony Brook University, Stony Brook, USA
l.lt@mail.scut.edu.cn, heng.fan@unt.edu, zhipeng.zhang.cv@outlook.com
yxu@scut.edu.cn, hling@cs.stonybrook.edu

## Abstract

Recently Transformer has been largely explored in tracking and shown state-of-the-art (SOTA) performance. However, existing efforts mainly focus on fusing and enhancing features generated by convolutional neural networks (CNNs). The potential of Transformer in representation learning remains under-explored. In this paper, we aim to further unleash the power of Transformer by proposing a simple yet efficient fully-attentional tracker, dubbed **SwinTrack**, within classic Siamese framework. In particular, both representation learning and feature fusion in SwinTrack leverage the Transformer architecture, enabling better feature interactions for tracking than pure CNN or hybrid CNN-Transformer frameworks. Besides, to further enhance robustness, we present a novel motion token that embeds historical target trajectory to improve tracking by providing temporal context. Our motion token is lightweight with negligible computation but brings clear gains. In our thorough experiments, SwinTrack exceeds existing approaches on multiple benchmarks. Particularly, on the challenging LaSOT, SwinTrack sets a new record with **0.713** SUC score. It also achieves SOTA results on other benchmarks. We expect SwinTrack to serve as a solid baseline for Transformer tracking and facilitate future research. Our codes and results are released at `https://github.com/LitingLin/SwinTrack`.

## 1   Introduction

Visual tracking has seen considerable progress since deep learning. In particular, the recent Transformer [30] has significantly pushed the state-of-the-art in tracking owing to its ability in modeling long-range dependencies. However, existing methods usually leverage Transformer for fusing and enhancing features generated from convolutional neural networks (CNNs), *e.g.*, ResNet [14]. The potential of exploiting Transformer for feature representation learning is largely under-explored.

Recently, Vision Transformer (ViT) [7] has exhibited great potential in robust feature representation learning. Particularly, its extension Swin Transformer [23] has achieved state-of-the-art (SOTA) results on multiple tasks. Taking inspiration from this, we argue, besides the feature fusion, the representation learning in tracking can also benefit from Transformer via attention. Thus motivated, we propose to develop a fully attentional tracking framework based on Siamese architecture. Specifically, both the feature representation learning and the feature fusion of template and search region are realized by Transformer. More concretely, we borrow the architecture of the powerful

---

[*]Equal Contributions.

36th Conference on Neural Information Processing Systems (NeurIPS 2022).

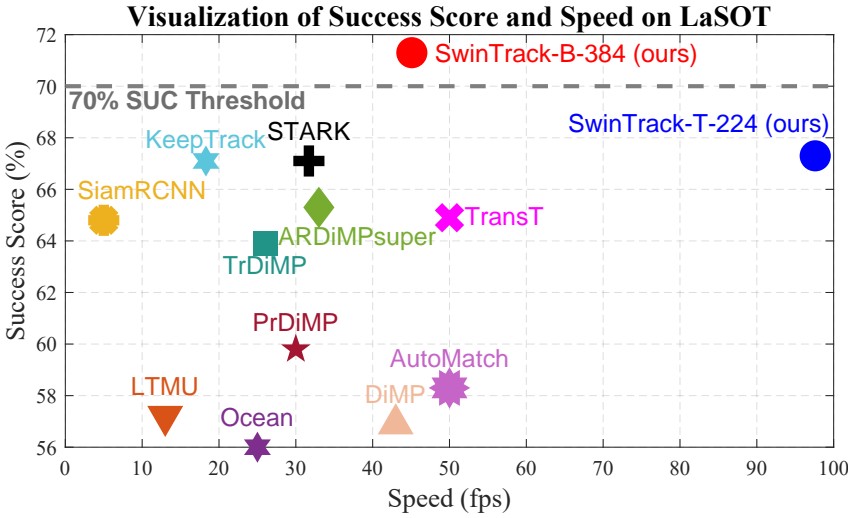

Figure 1: Comparison on LaSOT [9]. Our tracker (SwinTrack-B-384) sets a new record with 0.713 SUC score and still runs efficiently at around 45 *fps*. A lighter version (SwinTrack-T-224) achieves 0.672 SUC score and runs at around 96 *fps*, which is on par with existing SOTAs in accuracy but much faster.

Swin Transformer [23] and adapt it to Siamese tracking. Note that, other Transformer architectures can be used. For feature fusion, we introduce a simple homogeneous concatenation-based fusion architecture, without a query-based decoder.

Moreover, taking into consideration that tracking is a temporal task, we propose a novel motion token to improve robustness. Inspired by that the target usually moves smoothly in a short period, motion token is represented by the historical target trajectory within a local temporal window. We incorporate the (single) motion token in the decoder of feature fusion to leverage motion information during tracking. Despite being conceptually simple, our motion token can effectively boost tracking performance, with negligible computation.

We name our framework SwinTrack. As a pure Transformer framework, SwinTrack enables better interactions inside the feature learning of template and search region and their fusion compared to pure CNN-based [1, 20] and hybrid CNN-Transformer [5, 32, 36] frameworks, leading to more robust performance (see Fig. 1). Fig. 2 demonstrates the architecture of SwinTrack. We conduct extensive experiments on five large-scale benchmarks to verify the effectiveness of SwinTrack, including LaSOT [9], LaSOT$_{ext}$ [8], TrackingNet [26], GOT-10k [15] and TNL2k [34]. On all benchmarks, SwinTrack achieves promising results and meanwhile runs fast at 45 *fps*. In particular, on the challenging LaSOT, SwinTrack sets a new record of 71.3 SUC score, surpassing the strongest prior tracker [36] (to date) by 3.1 absolute percentage points and crossing the 0.7 SUC threshold *for the first time* (see Fig. 1 again). It also achieves 49.1 SUC, 84.0 SUC, 72.4 AO and 55.9 SUC scores on LaSOT$_{ext}$, TrackingNet, GOT-10k and TNL2k respectively, which are better than or on par with state-of-the-arts (SoTAs). In addition, we provide a lighter version of SwinTrack that obtains comparable results to SoTAs but runs much faster at around 98 *fps*.

In summary, our contributions are as follows: (**i**) We propose SwinTrack, a simple and strong baseline for fully attentional tracking; (**ii**) We present a simple yet effective motion token, enabling the integration of rich motion context during tracking, further boosting the robustness of SwinTrack, with negligible computation; (**iii**) Our proposed SwinTrack achieves state-of-the-art performance on multiple benchmarks. We believe SwinTrack further shows the potential of Transformer and expect it to serve as a baseline for future research.

## 2 Related Work

**Siamese Tracking.** The Siamese tracking methods formulate tracking as a matching problem and aim to offline learn a generic matching function for this task. The seminal method of [1] introduces

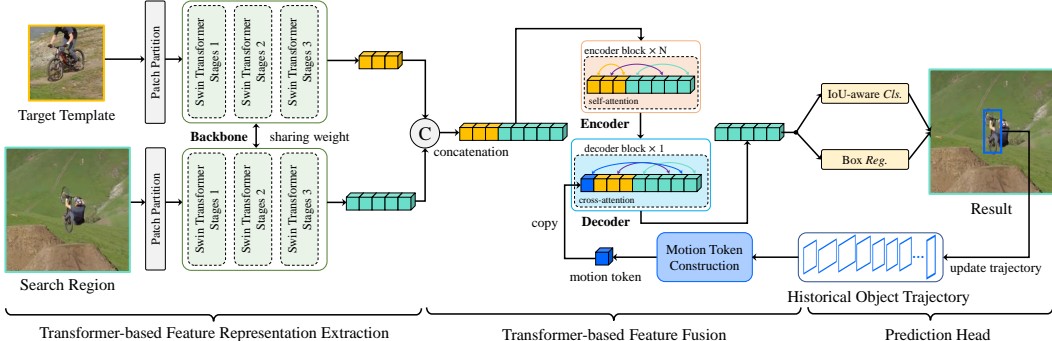

Figure 2: Architecture of SwinTrack, which contains three parts including Transformer-based feature representation extraction, Transformer-based feature fusion and prediction head. Our SwinTrack is a simple and neat tracking framework without complex designs such as multi-scale features or temporal template updating, yet demonstrating state-of-the-art performance. *Best viewed in color.*

a fully convolutional Siamese network for tracking and shows a good balance between accuracy and speed. To improve Siamese tracking in handling scale variation, the work of [20] incorporates the region proposal network (RPN) [27] into the Siamese network and proposes the anchor-based tracker, showing higher accuracy with faster speed. Later, numerous extensions have been presented to improve Siamese tracking, including deeper backbone [19], multi-stage architecture [10, 11], anchor-free Siamese trackers [41], deformable attention [37], to name a few.

**Transformer in Vision.** Transformer [30] originates from natural language processing (NLP) for machine translation and has been introduced to vision recently and shows great potential. The work of [3] first uses Transformer for object detection and achieved promising results. To explore the capability of Transformer in representation learning, the work of [7] applies Transformer to construct backbone network, and the resulting Vision Transformer (ViT) attains excellent performance compared to convolutional networks while requiring fewer training resources, which encourages many extensions upon ViT[29, 4, 38, 33, 23]. Among them, the Swin Transformer [23] has received extensive attention. It proposes a simple shifted window strategy to replace the fixed-patch method in ViT, which significantly improves efficiency and meanwhile demonstrates state-of-the-art results on multiple image tasks. Our work is inspired by Swin Transformer, but differently, we focus on the video task of visual tracking.

**Transformer in Tracking.** Inspired by the success in other fields, researchers have leveraged Transformer for tracking. The method of [5] applies Transformer to enhance and fuse features in the Siamese tracking for improvement. The approach of [32] uses Transformer to exploit temporal features to improve tracking robustness. The work of [36] introduces a new transformer architecture dedicated to visual tracking, explores the Spatio-temporal Transformer by integrating the model updating operations into a Transformer module.

Our SwinTrack is related to but significantly different from the above Transformer-based trackers. Specifically, the aforementioned methods mainly apply Transformer to fuse convolutional features and belong to the hybrid CNN-Transformer architecture. Unlike them, SwinTrack is a pure Transformer-based tracking architecture where both representation learning and feature fusion are realized with Transformer, enabling the exploration of better features for robust tracking.

## 3 Tracking via Vision-Motion Transformer

We present SwinTrack, a vision-motion integrated Transformer for object tracking, in Fig. 2. The proposed framework contains three main components, *i.e.*, the Swin-Transformer backbone for feature extraction, the encoder-decoder network for mixing vision-motion cues, and the head network for localizing targets. In the following sections, we first shortly describe the Swin-Transformer backbone network, then elaborate on the proposed vision-motion encoder-decoder. Afterward, we give a discussion about our method and shortly describe the network head and training loss.

### 3.1 Swin-Transformer for Feature Extraction

The deep convolutional neural network has significantly improved the performance of trackers. Along with the advancement of trackers, the backbone network has evolved twice: AlexNet [17] and ResNet [14]. Swin-Transformer [23], in comparison to ResNet, can give a more compact feature representation and richer semantic information to assist succeeding networks in better localizing the target objects (demonstrate in the ablation study demonstrated in the ablation study), which is thus chosen for basic feature extraction in our model.

Our tracker, following Siamese tracking framework [1], requires a pair of image patches as inputs, *i.e.*, template image $\mathbf{z} \in \mathbb{R}^{H_z \times W_z \times 3}$ and search region image $\mathbf{x} \in \mathbb{R}^{H_x \times W_x \times 3}$. As in the typical Swin-Transformer procedure, template and search region images are divided to non-overlapped patches and sent to the network, which generates template tokens (dubbed **T-tokens**) $\varphi(\mathbf{z}) \in \mathbb{R}^{\frac{H_z}{s} \frac{W_z}{s} \times C}$ and search region tokens (dubbed **S-tokens**) $\varphi(\mathbf{x}) \in \mathbb{R}^{\frac{H_x}{s} \frac{W_x}{s} \times C}$. $s$ is the stride of the backbone network. Since there is no dimension projection in our model, $C$ is the hidden dimension of the whole model.

### 3.2 Vision-Motion Representation Learning

The essential step for matching-based visual tracking is injecting the template information into the search region. In our framework, we adopt an encoder to fuse the features from the template and the search region, meanwhile, a decoder is arranged to achieve vision-motion representation learning, as illustrated in Fig. 2.

**Encoder for fusing template and search tokens.** The encoder contains a sequence of Transformer blocks where each consists of a multi-head self-attention (MSA) module and a feed-forward network (FFN). FFN contains a two-layers multi-layer perceptron (MLP), GELU activation layer is inserted after the first linear layer. Layer normalization (LN) is always conducted before every module (MSA and FFN). Residual connection is applied to MSA and FFN modules.

Before feeding the features into the encoder, the template and search region tokens are concatenated along spatial dimensions to generate a mixing representation $\mathbf{f}_m$. For each block, the MSA module computes self-attention over mixing union representation, which equals to separately conducting self-attention on T-tokens/S-tokens and meanwhile performing cross-attention between T-tokens and S-tokens, but more efficient. FFN refines the features generated by MSA. When the tokens get out of the encoder, a de-concatenation operation is arranged to decouple the template and search region tokens. The process of encoder can be expressed as:

$$\mathbf{f}_m^1 = \text{Concat}(\varphi(\mathbf{z}), \varphi(\mathbf{x}))$$
$$\cdots$$
$$\mathbf{f}_m^{l'} = \mathbf{f}_m^l + \text{MSA}(\text{LN}(\mathbf{f}_m^l))$$
$$\mathbf{f}_m^{l+1} = \mathbf{f}_m^{l'} + \text{FFN}(\text{LN}(\mathbf{f}_m^{l'})) \tag{1}$$
$$\cdots$$
$$\mathbf{f}_z^L, \mathbf{f}_x^L = \text{DeConcat}(\mathbf{f}^L),$$

where $l$ denotes the $l$-th layer and $L$ denotes the number of blocks.

**Decoder for fusing vision and motion information.** Before describing the architecture of decoder, we first detail how to generate a motion token (dubbed **M-token**). Motion token is the embedding of the historical trajectory of the target object. The past object trajectory is represented as a set of target object box coordinates, $T = \{\circledcirc_1, \circledcirc_2, ..., \circledcirc_t\}$, where $t$ represents the frame index, $\circledcirc$ is the bounding box of target object. $\circledcirc$ is defined by the top-left and bottom-right corners of the target object, denotes as $\circledcirc_t = (o_t^{x_1}, o_t^{y_1}, o_t^{x_2}, o_t^{y_2})$. For flexible modeling, a sampling process is required to ensure the following properties: **1) fixed length**, **2) focusing on the latest trajectories** and **3) reducing redundancy**. In our method, we sample object trajectory as:

$$\mathcal{T} = \{\circledcirc_{s(1)}, \circledcirc_{s(2)}, ..., \circledcirc_{s(n)}\}, \quad \text{where } s(i) = max(t - i \times \Delta, 1), \tag{2}$$

$n$ is the number of sampled object trajectories, $\Delta$ is the fixed sampling interval. For Siamese tracker, the search region is cropped from the input image. In detail, a cropping with resizing operation can be

used to describe the process. Giving the point in the input image as $(\mathbf{x}_i, \mathbf{y}_i)$, the corresponding point in the search region as $(\mathbf{x}_o, \mathbf{y}_o)$, we can formulate the cropping process employed in pre-processing of the Siamese Tracker as $\mathbf{x}_o = (\mathbf{x}_i - i_x)s_x + o_x$ and $\mathbf{y}_o = (\mathbf{y}_i - i_y)s_y + o_y$, where $(i_x, i_y)$ is the center of the cropping window in the input image, $(s_x, s_y)$ is the scaling factor, $(o_x, o_y)$ is the center of cropped and scaled window in the search region. We apply the same transformation on the sampled object trajectory to make the object trajectory invariant to the cropping, denoting $\bar{\mathcal{T}} = \{\bar{\mathbb{O}}_{s(1)}, \bar{\mathbb{O}}_{s(2)}, ..., \bar{\mathbb{O}}_{s(n)}\}$ as the result.

Then, to embed the transformed object trajectory into the network, we adopt four embedding matrices to embed the elements in box coordinates separately. We denotes the embedding matrix as $W \in \mathbb{R}^{(\mathrm{g}+1) \times d}$, $\mathrm{g}$ controls the embedding granularity of the object trajectory, $d$ is the size of each embedding vector. The last entry of the embedding matrix is used as the padding vector, indicating an invalid state, like object absence or out of the search region. Thus, we normalize the sampled target object box coordinates in the range $[1, \mathrm{g}]$, and quantize to integers to get the index of embedding vector:

$$
\begin{aligned}
\hat{\mathcal{T}} &= \{\hat{\mathbb{O}}_{s(1)}, \hat{\mathbb{O}}_{s(2)}, ..., \hat{\mathbb{O}}_{s(n)}\}, \\
\text{where } \hat{\mathbb{O}}_{s(i)} &= [\mathrm{n}(\bar{\mathbb{O}}^{x_1}_{s(i)}, w), \mathrm{n}(\bar{\mathbb{O}}^{y_1}_{s(i)}, h), \mathrm{n}(\bar{\mathbb{O}}^{x_2}_{s(i)}, w), \mathrm{n}(\bar{\mathbb{O}}^{y_2}_{s(i)}, h)], \\
\mathrm{n}(o, l) &= \begin{cases} \lfloor \frac{o}{l} \times \mathrm{g} \rfloor & \text{if valid,} \\ \mathrm{g} + 1 & \text{else,} \end{cases}
\end{aligned}
\tag{3}
$$

$(w, h)$ is the size of search region feature map.

Finally, the motion token $\mathbf{E}_{motion} \in \mathbb{R}^{1 \times d}$ is given by a concatenation of all box coordinate embedding of the sampled object trajectory. FLOPs is negligible because the construction of motion token is just a composition of embedding lookups and token concatenation.

The decoder consists of a multi-head cross-attention(MCA) module and a feed-forward network(FFN). The decoder takes the outputs from the encoder and the motion token as input, generating the final vision-motion representation $\mathbf{f}_{vm} \in \mathbb{R}^{\frac{H_x}{s} \times \frac{W_x}{s} \times C}$ of by computing cross-attention over $\mathbf{f}_x^L$ and $\mathrm{Concat}(\mathbf{E}_{motion}, \mathbf{f}_z^L, \mathbf{f}_x^L)$. The decoder is akin to a layer in the encoder, except that the correlation between the template tokens and the search tokens is dropped since we do not need to update the features from the template image in the last layer. The process of the decoder is formulated as:

$$
\begin{aligned}
\mathbf{f}_m^D &= \mathrm{Concat}(\mathbf{E}_{motion}, \mathbf{f}_z^L, \mathbf{f}_x^L) \\
\mathbf{f}'_{vm} &= \mathbf{f}_x^L + \mathrm{MCA}(\mathrm{LN}(\mathbf{f}_x^L), \mathrm{LN}(\mathbf{f}_m^D)) \\
\mathbf{f}_{vm} &= \mathbf{f}'_{vm} + \mathrm{FFN}(\mathrm{LN}(\mathbf{f}'_{vm})).
\end{aligned}
\tag{4}
$$

$\mathbf{f}_{vm}$ will feed to the head network to generate a classification response map and a bounding box regression map.

**Positional encoding.** Transformer requires a positional encoding to identify the position of the current processing token[30] because the self-attention module is permutation-invariance. We adopt the *untied positional encoding* [16] as our positional encoding method. The *untied positional encoding* enhances the expressiveness of the model through untie the positional embeddings from token embeddings with an isolated positional embedding matrix. It also considers the case of special tokens, like the motion token in this paper. We generalize the *untied positional encoding* to multi-dimensions multi-sources data to comply with *concatenated-based fusion* in our tracker. See the appendix for the details.

### 3.3 Discussion

**Why concatenated attention?** To simplify the description, we call the method described above *concatenation-based fusion*. To fuse and process features from multiple sources, it is intuitive to perform self-attention on the feature from each source separately and then compute cross-attention across features from different sources. We call this method *cross-attention-based fusion*. Transformer makes fewer assumptions about the spatial structure of data, which provides great modeling flexibility.

In comparison to *cross-attention-based fusion*, *concatenation-based fusion* can save computation cost through operation sharing and reduce model parameters through weight sharing. From the perspective of metric learning, weight sharing is an essential design to ensure the metric between two branches of data is symmetric. Through *concatenation-based fusion*, we implement this property not only in the feature extraction stage but also in the feature fusion stage. In general, *concatenation-based fusion* improves both efficiency and performance.

**Why not window-based self/cross-attention?** Since we select stage 3 of the Swin-Transformer as the output, the number of tokens involved is significantly reduced, window-based attention cannot save too many FLOPs. Furthermore, considering the extra latency introduced by the window partition and window reverse operations, window-based attention may even be the slower one.

**Why not a query-based decoder?** Derivated from vanilla Transformer decoder, many transformer-based models in vision tasks leverage a learnable query to extract the desired objective features from the encoder, like object queries in [3], target query in [36]. However, in our experiment, a query-based decoder suffers from slow convergence and inferior performance. Most Siamese trackers [20, 35, 13] formulate tracking as a foreground-background classification problem, which can better exploit the background information. The vanilla Transformer decoder is a generative model, the generative approaches are considered not suitable for the classification tasks. In another aspect, learning a general target query for any kind of object might cause a bottleneck. In terms of vanilla Transformer encoder-decoder architecture, SwinTrack is an "encoder" only model. Furthermore, quite a little domain knowledge can be easily applied on a classic Siamese tracker to improve the performance, like introducing the smooth movement assumption by using Hanning penalty window on the response map.

**Are other forms of motion token feasible?** Other forms to construct motion token are possible, such as constructing motion token by summing up the past box coordinate embeddings or representing past object trajectories by one token per box. In our early experiments, we find that the proposed motion token is more effective with the best performance. Summing up the past box coordinate embeddings may result in over-parameterization on the coordinate embeddings. While adding temporal motion representation along with visual features to the single-layer decoder in a multi-token form is ineffective, precise temporal modeling may be required in this form.

### 3.4 Head and Loss

**Head.** The head network is split into two branches: classification and bounding box regression. Each of them is a three-layer perceptron. And both of them receives the feature map from the decoder as input to predict the classification response map $r_{cls} \in \mathbb{R}^{(H_x \times W_x) \times 1}$ and bounding box regression map $r_{reg} \in \mathbb{R}^{(H_x \times W_x) \times 4}$, respectively.

**Classification loss.** In classification branch, we employ the *IoU-aware classification score* as the training target and the *varifocal loss* [39] as the training loss function. IoU-aware design has been very popular recently, but most works consider IoU prediction as an auxiliary branch to assist classification or bounding box regression [41, 2, 35]. To remove the gap between different prediction branches, [39] and [21] replace the hard classification target from the ground-truth value, (i.e., 1 for positive samples, 0 for negative samples), to the IoU between the predicted bounding box and the ground-truth one, which is named the *IoU-aware classification score* (IACS). IACS can help the model select a more accurate bounding box prediction candidate from the pool by trying to predict the quality of the bounding box prediction in another branch at the same position. Along with the IACS, the varifocal loss was proposed in [39] to help the IACS approach outperform other IoU-aware designs.

The classification loss can be formulated as:

$$\mathbb{L}_{cls} = \mathbb{L}_{\text{VFL}}(p, \text{IoU}(b, \hat{b})), \tag{5}$$

where $p$ denotes the predicted IACS, $b$ denotes the predicted bounding box, and $\hat{b}$ denotes the ground-truth bounding box.

**Regression loss.** For bounding box regression, we employ the generalized IoU loss[28]. The regression loss function can be formulated as:

$$\mathbb{L}_{reg} = \sum_j \mathbb{1}_{\{\text{IoU}(b_j, \hat{b}) > 0\}} [p \mathbb{L}_{\text{GIoU}}(b_j, \hat{b})]. \tag{6}$$

The GIoU loss is weighted by $p$ to emphasize the high classification score samples. The training signals from the negative samples are ignored.

## 4 Experiments

### 4.1 Implementation

**Model.** We design two variants of SwinTrack with different configurations as follows:

- **SwinTrack-T-224**.
  Backbone: Swin Transformer-Tiny [23], pretrained with ImageNet-1k;
  Template size: $[112 \times 112]$; Search region size: $[224 \times 224]$; $C = 384$; $N = 4$;
- **SwinTrack-B-384**.
  Backbone: Swin Transformer-Base [23], pretrained with ImageNet-22k;
  Template size: $[192 \times 192]$; Search region size: $[384 \times 384]$; $C = 512$; $N = 8$;

where $C$ and $N$ are the channel number of the hidden layers in the first stage of Swin Transformer and the number of encoder blocks in feature fusion, respectively. In all variants, we use the output after the third stage of Swin Transformer for feature extraction. Thus, the backbone stride $s$ is 16.

For motion token, the number of sampled object trajectory $n$ is set to 16, the fixed sampling interval $\Delta$ is set to 15. If the frame rate of the video sequence is available, the sampling interval is adjusted according to the frame rate. Suppose the frame rate is $\mathbb{f}$, the new sampling interval is getting by $\frac{\Delta}{30}\mathbb{f}$, 30 fps is the standard frame rate we assumed. $\mathbb{g}$, which controls the embedding granularity, is set to the same size as the search region feature map, like 14 for SwinTrack-T-224, and 24 for SwinTrack-B-384. For the model for GOT-10k sequences, $n$ is set to 8, $\Delta$ is set to 8, and no frame rate adjustment is applied.

**Training.** We train SwinTrack using the training splits of LaSOT [9], TrackingNet [26], GOT-10k [15] (1,000 videos are removed following [36] for fair comparison) and COCO 2017 [22]. In addition, we report the results of SwinTrack-T-224 and SwinTrack-B-384 with GOT-10k training split only to follow the protocol described in [15].

The model is optimized with AdamW [24], with a learning rate of 5e-4, and a weight decay of 1e-4. The learning rate of the backbone is set to 5e-5. We train the network on 8 NVIDIA V100 GPUs for 300 epochs with 131,072 samples per epoch. The learning rate is dropped by a factor of 10 after 210 epochs. A 3-epoch linear warmup is applied to stabilize the training process. DropPath [18] is applied on the backbone and the encoder with a rate of 0.1. For the models trained for the GOT-10k evaluation protocol, to prevent over-fitting, the training epoch is set to 150, and the learning rate is dropped after 120 epochs.

For the motion token, the object trajectory for the Siamese training pair is generated with the method described above. The frames that object annotated as absent or out of the video sequence are marked as invalid, the corresponding box coordinates set to $-\infty$. Since the coarse granularity of the coordinate embedding in our setting is already can be seen as an augmentation of historical object trajectory, no additional data augmentation is applied.

**Inference.** We follow the common procedures for Siamese network-based tracking [1]. The template image is cropped from the first frame of the video sequence. The target object is in the center of the image with a background area factor of 2. The search region is cropped from the current tracking frame, and the image center is the target center position predicted in the previous frame. The background area factor for the search region is 4.

Our SwinTrack takes the template image and search region as inputs and output classification map $r_{cls}$ and regression map $r_{reg}$. To utilize positional prior in tracking, we apply hanning window penalty on $r_{cls}$, and the final classification map $r'_{cls}$ is obtained via $r'_{cls} = (1 - \gamma) \times r_{cls} + \gamma \times h$, where $\gamma$ is the weight parameter and $h$ is the Hanning window with the same size as $r_{cls}$. The target position is determined by the largest value in $r'_{cls}$ and the scale is estimated based on the corresponding regression results in $r_{reg}$.

For the motion token, the predicted confidence score and bounding box are collected on the fly. A confidence threshold $\theta_{conf}$ is applied, if the confidence score given by the classification branch of the

Table 1: Experiments and comparisons on five benchmarks: LaSOT, LaSOT$_{ext}$, TrackingNet, GOT-10k and TNL2k.

| Tracker | LaSOT [9] | | LaSOT$_{ext}$ [8] | | TrackingNet [26] | | GOT-10k [15] | | | TNL2k [34] | |
| --- | --- | --- | --- | --- | --- | --- | --- | --- | --- | --- | --- |
| | SUC | P | SUC | P | SUC | P | AO | SR$_{0.5}$ | SR$_{0.75}$ | SUC | P |
| C-RPN [10] | 45.5 | 44.3 | 27.5 | 32.0 | 66.9 | 61.9 | - | - | - | - | - |
| SiamPRN++ [19] | 49.6 | 49.1 | 34.0 | 39.6 | 73.3 | 69.4 | 51.7 | 61.6 | 32.5 | 41.3 | 41.2 |
| Ocean [41] | 56.0 | 56.6 | - | - | - | - | 61.1 | 72.1 | 47.3 | 38.4 | 37.7 |
| DiMP [2] | 56.9 | 56.7 | 39.2 | 45.1 | 74.0 | 68.7 | 61.1 | 71.7 | 49.2 | 44.7 | 43.4 |
| LTMU [6] | 57.2 | 57.2 | 41.4 | 47.3 | - | - | - | - | - | 48.5 | 47.3 |
| SiamR-CNN [31] | 64.8 | - | - | - | 81.2 | 80.0 | 64.9 | 72.8 | 59.7 | 52.3 | 52.8 |
| STMTrack [12] | 60.6 | 63.3 | - | - | 80.3 | 76.7 | 64.2 | 73.7 | 57.5 | - | - |
| AutoMatch [40] | 58.3 | 59.9 | 37.6 | 43.0 | 76.0 | 72.6 | 65.2 | 76.6 | 54.3 | - | - |
| TrDiMP [32] | 63.9 | 61.4 | - | - | 78.4 | 73.1 | 67.1 | 77.7 | 58.3 | - | - |
| TransT [5] | 64.9 | 69.0 | - | - | 81.4 | 80.3 | 67.1 | 76.8 | 60.9 | 51.0 | - |
| STARK [36] | 67.1 | - | - | - | 82.0 | - | 68.8 | 78.1 | 64.1 | - | - |
| KeepTrack [25] | 67.1 | 70.2 | 48.2 | - | - | - | - | - | - | - | - |
| SwinTrack-T-224 | 67.2 | 70.8 | 47.6 | 53.9 | 81.1 | 78.4 | 71.3 | 81.9 | 64.5 | 53.0 | 53.2 |
| SwinTrack-B-384 | 71.3 | 76.5 | 49.1 | 55.6 | 84.0 | 82.8 | 72.4 | 80.5 | 67.8 | 55.9 | 57.1 |

head is lower than the threshold, the target object in the current frame is marked as lost by setting the collected bounding box to $-\infty$. $\theta_{conf}$ is set to $0.4$ for LaSOT, the rests are set to $0.3$.

## 4.2 Comparisons to State-of-the-arts

We conduct experiments and compare SwinTrack with SoTA trackers on five benchmarks.

**LaSOT.** LaSOT [9] consists of 280 videos for test. Tab. 1 shows the results and comparisons with SoTAs. From Tab. 1, we can observe that SwinTrack-T-224 with light architecture reaches SoTA performance with 0.672 SUC and 0.708 PRE scores, which is competitive compared with other Transformer-based trackers, including STARK-ST101 (0.671 SUC score) and TransT (0.649 SUC), and other trackers using complicated designs such as KeepTrack (0.671 SUC) and SiamR-CNN (0.648 SUC score). With a larger backbone and input size, our strongest variant SwinTrack-B-384 sets a new record with 0.713 SUC score, surpassing START-ST101 and KeepTrack by 4.2 absolute percentage points.

**LaSOT$_{ext}$.** The recent LaSOT$_{ext}$ [8] is an extension of LaSOT by adding 150 extra videos. These new sequences are challenging as many similar distractors cause difficulties for tracking. The results of our tracker related to this dataset are an average of three times. KeepTrack uses a complex association technique to handle distractors and achieves a promising 0.482 SUC score as in Tab. 1. Compared with complicated KeepTrack, SwinTrack-T-224 is simple and neat, yet shows comparable performance with 0.476 SUC score. In addition, due to complicated design, KeepTrack runs at less than 20 *fps*, while SwinTrack-T-224 runs in 98 *fps*, 5× faster than KeepTrack. When using a larger model, SwinTrack-B-384 shows the best performance with 0.491 SUC score.

**TrackingNet.** We evaluate different trackers on the test set of TrackingNet [26]. From Tab. 1, we observe that our SwinTrack-T-224 achieves a comparable result with 0.811 SUC score. Using a larger model and input size, SwinTrack-B-384 obtains the best performance with 0.840 SUC score, better than STARK-ST101 with 0.820 SUC score and TransT with 0.814 SUC score.

**GOT-10k.** GOT-10k [15] offers 180 videos for test and it requires trackers to be trained using GOT-10k train split only. From Tab. 1, we see that SwinTrack-B-384 achieves the best mAO of 0.724, and SwinTrack-T-224 obtains a mAO of 0.713. Both models outperform other Transformer-based counterparts significantly, including START-ST101 (0.688 mAO), TransT (0.671 mAO) and TrDiMP (0.671 mAO).

**TNL2k.** TNL2k [34] is a newly released tracking dataset with 700 videos for test. As reported in Tab. 1, both models surpass the others. SwinTrack-B-384 set a new state-of-the-art with 0.559 SUC score.

---

[2] Multiply–accumulate operation

Table 2: Comparison on running speed and # parameters with other Transformer-based trackers.

| Tracker | Speed (*fps*) | MACs[2] (G) | Params (M) |
|---|---|---|---|
| TrDiMP [32] | 26 | - | - |
| TransT [5] | 50 | - | 23 |
| STARK-ST50 [36] | 42 | 10.9 | 24 |
| STARK-ST101 [36] | 32 | 18.5 | 42 |
| SwinTrack-T-224 | 98 | 6.4 | 23 |
| SwinTrack-B-384 | 45 | 69.7 | 91 |

Table 3: Ablation experiments of SwinTrack on four benchmarks. The experiments are conducted on SwinTrack-T-224 without the motion token. ❶: baseline method, *i.e.*, SwinTrack-T-224 without motion token; ❷: replacing Transformer backbone in the baseline method with ResNet-50; ❸: replacing our feature fusion with cross attention-based fusion in the baseline method; ❹: replacing the decoder in baseline with a target query-based; ❺: replacing united positional encoding with absolute sine position encoding in the baseline method; ❻: replacing the IoU-aware classification loss with the plain binary cross entropy loss; ❼: removing the Hanning penalty window in the baseline method inference.

| | LaSOT SUC (%) | LaSOT$_{ext}$ SUC (%) | TrackingNet SUC (%) | GOT-10k[3] mAO (%) | Speed *fps* | Params M |
|---|---|---|---|---|---|---|
| ❶ | 66.7 | 46.9 | 80.8 | 70.9 | 98 | 22.7 |
| ❷ | 64.2 | 41.8 | 79.5 | 68.2 | 121 | 20.0 |
| ❸ | 66.6 | 45.4 | 80.2 | 69.3 | 72 | 34.6 |
| ❹ | 66.6 | 43.2 | 79.6 | 69.0 | 91 | 25.3 |
| ❺ | 65.7 | 45.0 | 80.0 | 70.0 | 103 | 21.6 |
| ❻ | 66.2 | 46.7 | 79.4 | 68.2 | 98 | 22.7 |
| ❼ | 65.7 | 46.0 | 80.0 | 69.6 | 98 | 22.7 |

**Efficiency comparison.** We report the comparisons of SwinTrack with other Transformer-based trackers in terms of efficiency and complexity. As displayed in Tab. 2, SwinTrack-T-224 with a small model runs the fastest with a speed of *98 fps*. Especially, compared with STARK-ST101 and STARK-ST50 with 32 *fps* and 42 *fps*, SwinTrack-T-224 is 3× and 2× faster. Despite using a larger model, our SwinTrack-B-384 is still faster than STARK-ST101 and STARK-ST50.

### 4.3 Ablation Experiment

**Comparison with ResNet backbone.** We compare the Swin-Transformer backbone with popular ResNet-50 [14]. As shown in Tab. 3 (❶ *vs.* ❷). The Swin Transformer backbone significantly boosts the performance by 2.5% SUC score in LaSOT, 5.1% SUC score in LaSOT$_{ext}$. The result shows that the strong appearance modeling capability provided by the Swin Transformer plays a crucial role.

**Feature fusion.** As displayed in Tab. 3 (❶ *vs.* ❸), compared with the *concatenation-based fusion*, the *cross attention-based fusion* runs at a slower speed, occupies much more memory, and also has an inferior performance on all datasets. Slower speed can be due to the latency brought by the extra operations. The parameter-sharing strategy not only just reduces the number of parameters but also benefits metric learning.

**Comparison with the query-based decoder.** Queries is commonly adopted in the decoder of Transformer network in vision tasks, e.g. object query [3] and target query [36]. Nevertheless, our empirical results in Tab. 3 (❶ *vs.* ❹) show that a target query-based decoder degrades the tracking performance on all benchmarks, even with 2× training pairs. As discussed, one possible reason is the generative model is not suitable for classification. Besides, learning a general target query for any kind of object may also be difficult.

**Position encoding.** We compare the united positional encoding used in SwinTrack and the original absolute position encoding in Transformer [30]. Notice, We make a little modification to the original absolute position encoding: Except for the 2D embedding, the index of token source (e.g. 1 for the

---

[3]The GOT-10k results in this column are trained with full training datasets.

Table 4: Ablation experiments on our proposed motion token on the tracking performance on four benchmarks. The experiments are conducted on SwinTrack-T-224. ❶: SwinTrack-T-224; ❷: SwinTrack-B-384; ❸: SwinTrack-T-224 without motion token; ❹: SwinTrack-B-384 without motion token; ❺: replacing the motion token in SwinTrack-T-224 with a learnable embedding token.

|  | LaSOT SUC (%) | LaSOT$_{ext}$ SUC (%) | TrackingNet SUC (%) | GOT-10k mAO (%) | Speed $fps$ |
|---|---|---|---|---|---|
| ❶ | 67.2 | 47.6 | 81.1 | 71.3 | 96 |
| ❷ | 71.3 | 49.1 | 84.0 | 72.4 | 45 |
| ❸ | 66.7 | 47.0 | 80.8 | 70.0 | 98 |
| ❹ | 70.2 | 48.5 | 84.0 | 70.7 | 45 |
| ❺ | 66.3 | 45.2 | 81.2 | 70.0 | 96 |

tokens from the template patch, 2 for the tokens from the search region patch) is also embedded. As shown in Tab. 3 (❶ *vs.* ❺), our method with united positional encoding obtains improvements with 0.8-1.9 absolute percentage points on the benchmarks with negligible loss in speed (98 *vs.* 103).

**Loss function.** From Tab. 3 (❶ *vs.* ❻), we observe that the model trained with varifocal loss significantly outperforms the one with binary cross entropy (BCE) loss without loss of efficiency. This result indicates that the varifocal loss can assist the classification branch of the head to generate an IoU-aware response map, and thus help the tracker to improve the tracking performance.

**Post processing.** One may wonder with highly discriminative Transformer architecture and IoU-aware classification loss does the hanning penalty window is still functional, which introduces a strong smooth movement assumption. In the experiments, we remove the hanning penalty window in post-processing, as shown in Tab. 3 (❶ *vs.* ❼), the performance is dropped by 1.0 SUC for LaSOT, 1.3 AO for GOT-10k in absolute percentage, and less than 1% in the SUC metric of other datasets. This suggests that the strong smooth movement assumption is still applicable for our tracker. But compared with the former Transformer-based tracker [5], the performance gap between with and without penalty window post-processing is narrowing.

**Effectiveness of motion token.** We study the effectiveness of the motion token by conducting comparison experiments. As shown in Tab. 4 (❶ vs. ❸ and ❷ vs. ❹), the models with motion token outperforms the models without motion token on all datasets, especially on LaSOT$_{ext}$ and GOT-10k. The results indicate that the motion token can assist the tracker to handle hard similar distractors in LaSOT$_{ext}$ and stabilize the short-term tracking like the sequences in GOT-10k test set. We also study whether the effectiveness of the motion token is simply from the extra embedding vector. We set up an experiment as in Tab. 4 (❺), which replaces the motion token with a learnable embedding token. The result shows that the extra embedding vector has negative impacts indicating the effectiveness of the embedding of object trajectory.

# 5 Conclusion

In this work, we present SwinTrack, a simple and strong baseline for Transformer tracking. In SwinTrack, both representation learning and feature fusion are implemented with the attention mechanism. Extensive experiments demonstrate the effectiveness of such architecture. Besides, we propose the motion token to enhance the robustness of the tracker by providing the historical object trajectory, showing the flexibility of the Transformer model in architectural design. With the power of sequence-to-sequence model architecture, a context-rich tracker is possible, and more contextual cues can be incorporated. Finally, We hope this work can inspire and facilitate future research.

## Acknowledgments and Disclosure of Funding

This work is supported by Peng Cheng Laboratory Research Project No. PCL2021A07. Heng Fan and his employer receive no financial support for the research, authorship, and/or publication of this article.

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
