# Appendix

## 1    Positional Encoding

Transformer requires a positional encoding to identify the position of the current processing token [17]. Through a series of comparison experiments, we choose *untied positional encoding*, which is proposed in TUPE [10], as the positional encoding solution of our tracker. In addition, we generalize the *untied positional encoding* to arbitrary dimensions to fit with other components in our tracker.

The original transformer [17] proposes a absolute positional encoding method to represent the position: a fixed or learnable vector $p_i$ is assigned to each position $i$. Starting from the basic attention module, we have:

$$\text{Atten}(Q, K, V) = \text{softmax}\Big(\frac{QK^T}{\sqrt{d_k}}V\Big), \tag{1}$$

where $Q$,$K$,$V$ are the *query* vector, *key* vector and *value* vector, which are the parameters of the attention function, $d_k$ is the dimension of *key*. Introducing the linear projection matrix and multi-head attention to the attention module (1), we get the multi-head variant defined in [17]:

$$\text{MultiHead}(Q, K, V) = \text{Concat}(\text{head}_1, ..., \text{head}_{\text{h}})W_O, \tag{2}$$

where $\text{head}_{\text{i}} = \text{Atten}(QW_i^Q, KW_i^K, VW_i^V)$, $W_i^Q \in \mathbb{R}^{d_{\text{model}} \times d_k}$, $W_i^K \in \mathbb{R}^{d_{\text{model}} \times d_k}$, $W_i^V \in \mathbb{R}^{d_{\text{model}} \times d_v}$, $W_i^O \in \mathbb{R}^{hd_v \times d_{\text{model}}}$ and $h$ is the number of heads. For simplicity, as in [10], we assume that $d_k = d_v = d_{\text{model}}$, and use the single-head version of self-attention module. Denoting the input sequence as $x = x_1, x_2, \ldots, x_n$, where $n$ is the length of sequence, $x_i$ is the $i$-th token in the input data. Denoting the output sequence as $z = (z_1, z_2, \ldots, z_n)$. Self-attention module can be rewritten as

$$z_i = \sum_{j=1}^{n} \frac{\exp(\alpha_{ij})}{\sum_{j'=1}^{n} \exp(\alpha_{ij'})}(x_j W^V), \tag{3}$$

$$\text{where } \alpha_{ij} = \frac{1}{\sqrt{d}}(x_i W^Q)(x_j W^K)^T. \tag{4}$$

Obviously, the self-attention module is permutation-invariance. Thus it can not "understand" the order of input tokens.

**Untied absolute positional encoding.** By adding a learnable positional encoding [17] to the single-head self-attention module, we can obtain the following equation:

$$\begin{aligned}
\alpha_{ij}^{Abs} &= \frac{((w_i + p_i)W^Q)((w_j + p_j)W^K)^T}{\sqrt{d}} \\
&= \frac{(w_i W^Q)(w_j W^K)^T}{\sqrt{d}} + \frac{(w_i W^Q)(p_j W^K)^T}{\sqrt{d}} \\
&\quad + \frac{(p_i W^Q)(w_j W^K)^T}{\sqrt{d}} + \frac{(p_i W^Q)(p_j W^K)^T}{\sqrt{d}}.
\end{aligned} \tag{5}$$

The equation (5) is expanded into four terms: token-to-token, token-to-position, position-to-token, position-to-position. [10] discuss the problems that exist in the equation and proposes the *untied absolute positional encoding*, which unties the correlation between tokens and positions by removing the token-position correlation terms in equation (5), and using an isolated pair of projection matrices $U^Q$ and $U^K$ to perform linear transformation upon positional embedding vector. The following is the new formula for obtaining $\alpha_{ij}$ using the *untied absolute positional encoding* in the $l$-th layer:

$$\begin{aligned}
\alpha_{ij} &= \frac{1}{\sqrt{2d}}(x_i^l W^{Q,l})(x_j^l W^{K,l})^T \\
&\quad + \frac{1}{\sqrt{2d}}(p_i U^Q)(p_j U^K)^T.
\end{aligned} \tag{6}$$

where $p_i$ and $p_j$ is the positional embedding at position $i$ and $j$ respectively, $U^Q \in \mathbb{R}^{d \times d}$ and $U^K \in \mathbb{R}^{d \times d}$ are learnable projection matrices for the positional embedding vector. When extending

to the multi-head version, the positional embedding $p_i$ is shared across different heads, while $U^Q$ and $U^K$ are different for each head.

**Relative positional bias.** According to [16], relative positional encoding is a necessary supplement to absolute positional encoding. In [10], a relative positional encoding is applied by adding a relative positional bias to equation (6):

$$
\begin{aligned}
\alpha_{ij} = {} & \frac{1}{\sqrt{2d}}(x_i^l W^{Q,l})(x_j^l W^{K,l})^T \\
& + \frac{1}{\sqrt{2d}}(p_i U^Q)(p_j U^K)^T + b_{j-i},
\end{aligned}
\tag{7}
$$

where for each $j - i$, $b_{j-i}$ is a learnable scalar. The *relative positional bias* is also shared across layers. When extending to the multi-head version, $b_{j-i}$ is different for each head.

**Generalize to multiple dimensions.** Before working with our tracker's encoder and decoder network, we need to extend the *untied positional encoding* to a multi-dimensional version. One straightforward method is allocating a positional embedding matrix for every dimension and summing up all embedding vectors from different dimensions at the corresponding index to represent the final embedding vector. Together with *relative positional bias*, for an n-dimensional case, we have:

$$
\begin{aligned}
\alpha_{\underbrace{ij\dots}_{n},\underbrace{mn\dots}_{n}} = {} & \frac{1}{\sqrt{2d}}(x_{\underbrace{ij\dots}_{n}}W^Q)(x_{\underbrace{mn\dots}_{n}}W^K)^T \\
& + \frac{1}{\sqrt{2d}}[\underbrace{(p_i^1 + p_j^2 + \dots)}_{n} U^Q][\underbrace{(p_m^1 + p_n^2 + \dots)}_{n} U^K]^T \\
& + b_{\underbrace{m-i,\,n-j,\,\dots}_{n}}.
\end{aligned}
\tag{8}
$$

**Generalize to concatenation-based fusion.** In order to work with *concatenation-based fusion*, the *untied absolute positional encoding* is also concatenated to match the real position, the indexing tuple of *relative positional bias* now appends with a pair of indices to reflect the origination of *query* and *key* involved currently.

Take $l$-th layer in the encoder as the example:

$$
\begin{aligned}
\alpha_{ij,mn,g,h} = {} & \frac{1}{\sqrt{2d}}(x_{ij,g}^l W^{Q,l})(x_{mn,h}^l W^{K,l})^T \\
& + \frac{1}{\sqrt{2d}}[(p_{i,g}^1 + p_{j,g}^2)U_g^Q][(p_{m,h}^1 + p_{n,h}^2)U_h^K]^T \\
& + b_{m-i,n-j,g,h},
\end{aligned}
\tag{9}
$$

where $g$ and $h$ are the index of the origination of *query* and *key* respectively, for instance, 1 for the tokens from the template image, 2 for the tokens from the search image. The form in the decoder is similar, except that $g$ is fixed. In our implementation, the parameters of *untied positional encoding* are shared inside the encoder and the decoder, respectively.

## 2 The Effect of Pre-training Datasets

The two variants of our tracker, SwinTrack-T-224 and SwinTrack-B-384 are using different pre-training datasets, which are derived from the settings from Swin Transformer [12]. Specifically, SwinTrack-T-224 adopts ImageNet-1k and SwinTrack-B-384 adopts ImageNet-22k.

To analyze the effect of different pre-training datasets, we conduct an experiment on the performance of our tracker with different pre-training datasets. Other than the pre-training datasets, The experiment follows the same settings in the ablation study in the paper, the motion token is not used and the results on GOT-10k are trained on the full datasets as described in the paper. From Tab. 1, we can observe that, for smaller model SwinTrack-T-224 (23M # parameters), pre-training on ImageNet-22k brings small improvements on LaSOT (+0.6%) and TrackingNet (+0.4%) but degrades the performance on

GOT-10k (-1.4%). For larger model SwinTrack-B-384 (91M # parameters), pre-training on ImageNet-22k shows significant performance gains on LaSOT (+2.2%) and GOT-10k (+3.0%) but slightly degrades the result on TrackingNet (-0.6%). On LaSOT$_{ext}$, ImageNet-22k shows a performance degradation on smaller model SwinTrack-T-224 (-0.9%) and brings small improvements on larger model SwinTrack-B-384 (+0.2%).

Table 1: The effect of Imagenet-22k pre-training. The results are following the settings in the ablation study in the paper (motion token is not used and the result on GOT-10k is trained on the full dataset).

| Trackers | Pre-training | LaSOT [6] | | LaSOT$_{ext}$ [5] | | TrackingNet [15] | | GOT-10k [9] | | |
| | | SUC | P | SUC | P | SUC | P | AO | SR$_{0.5}$ | SR$_{0.75}$ |
|---|---|---|---|---|---|---|---|---|---|---|
| SwinTrack-T-224 | ImageNet-1k | 66.7 | 70.6 | 46.9 | 52.9 | 86.7 | 80.1 | 69.7 | 79.0 | 65.6 |
| SwinTrack-T-224 | ImageNet-22k | 67.3 | 71.7 | 46.0 | 51.7 | 81.2 | 78.9 | 69.5 | 78.9 | 65.5 |
| SwinTrack-B-384 | ImageNet-1k | 68.0 | 72.5 | 47.3 | 53.2 | 83.8 | 82.9 | 71.8 | 80.2 | 67.1 |
| SwinTrack-B-384 | ImageNet-22k | 70.2 | 75.3 | 47.5 | 53.3 | 86.9 | 80.1 | 70.2 | 80.7 | 65.4 |

Table 2: Performance comparisons with newly released Transformer-based Trackers on four benchmarks: LaSOT, LaSOT$_{ext}$, TrackingNet and GOT-10k.

| Tracker | Pre-training | LaSOT [6] | | LaSOT$_{ext}$ [5] | | TrackingNet [15] | | GOT-10k [9] | | |
| | | SUC | P | SUC | P | SUC | P | AO | SR$_{0.5}$ | SR$_{0.75}$ |
|---|---|---|---|---|---|---|---|---|---|---|
| STARK [20] | ImageNet-1k | 67.1 | - | - | - | 82.0 | - | 68.8 | 78.1 | 64.1 |
| SBT [18] | ImageNet-1k | 66.7 | 71.1 | - | - | - | - | 70.4 | 80.8 | 64.7 |
| ToMP [13] | ImageNet-1k | 68.5 | 73.5 | 45.9 | - | 81.5 | 78.9 | - | - | - |
| MixFormer [3] | ImageNet-22k | 70.1 | 76.3 | - | - | 83.9 | 83.1 | - | - | - |
| AiATrack [7] | ImageNet-1k | 69.0 | 73.8 | 47.7 | 55.4 | 82.7 | 80.4 | 69.6 | 80.0 | 63.2 |
| Unicorn [19] | ImageNet-1k | 68.5 | 74.1 | - | - | 83.0 | 82.2 | - | - | - |
| OSTrack [21] | MAE [8] | 71.1 | 77.6 | 50.5 | 57.6 | 83.9 | 83.2 | 73.7 | 83.2 | 70.8 |
| SwinTrack-T-224 | ImageNet-1k | 67.2 | 70.8 | 47.6 | 53.9 | 81.1 | 78.4 | 71.3 | 81.9 | 64.5 |
| SwinTrack-B-384 | ImageNet-22k | 71.3 | 76.5 | 49.1 | 55.6 | 84.0 | 82.8 | 72.4 | 80.5 | 67.8 |

## 3 Comparison with Newly Released Transformer-based Trackers

We compare our tracker with some newly released Transformer-based trackers, including STARK [20], SBT [18], ToMP [13], MixFormer [3], AiATrack [7], Unicorn [19], OSTrack [21] in Tab. 2 in four challenging benchmarks. The result shows our tracker is still competitive.

Fig. 1 and Fig. 2 show the success plot and the precision plot respectively. The comparison includes our SwinTrack-T-224, our SwinTrack-B-384, TransT[2], STARK[20], MixFormer[3], AiATrack[7] and ToMP[13]. Our tracker obtained the best performance on this benchmark. By looking into the curves of the figures, there is a significant advantage in the bounding box accuracy compared with other trackers due to our fully attentional architecture.

The success AUC score under different attributes of LaSOT [6] Test set in shown in Fig. 3. Fig. 3 indicates that our tracker has no obvious shortcomings except the viewpoint change.

## 4 Results on UAV123 and VOT Benchmark

In this section, we report the performance of the tracker on three additional benchmarks, including UAV123 [14], VOT2020 and VOT-STB2022 [11].

UAV123[14] is an aerial video dataset and benchmark for low-altitude UAV target tracking, containing 123 video sequences. Our tracker is on par with the state-of-the-art, AiATrack [7], on this benchmark. The results are shown in Tab. 3.

Finally, we evaluate our tracker on the two versions of the VOT Challenge: VOT2020 and VOT-STB2022. The VOT2020 dataset contains 60 videos with segmentation masks annotated. Since our tracker is a bounding box only method, we compare the results with the trackers that produce the bounding boxes as well. The result in Tab.4 shows that SwinTrack-T-224 has a better performance than the larger SwinTrack-B-384 on this benchmark.

In addition, We report the results on VOT-STB2022 in Tab.5. SwinTrack-T-224 has a better performance on VOT-STB2022 as well. No comparison is made since VOT-STB2022 is a newly released benchmark.

Table 3: Comparison to the state-of-the-arts on UAV123 [14] benchmark.

|  | Ocean [22] | DiMP [1] | TransT [2] | ToMP 50[13] | MixFormer 22k[3] | AiATrack [7] | SwinTrack T-224 | SwinTrack B-384 |
|---|---|---|---|---|---|---|---|---|
| AUC (%) | 62.1 | 65.3 | 69.1 | 69.0 | 70.4 | 70.6 | 68.8 | 70.5 |

Table 4: Comparison to the state-of-the-art bounding box only methods on VOT2020ST [11].

|  | ATOM [4] | DiMP [1] | STARK 50[20] | STARK 101[20] | ToMP 50[13] | ToMP 101[13] | SwinTrack T-224 | SwinTrack B-384 |
|---|---|---|---|---|---|---|---|---|
| EAO | 0.271 | 0.274 | 0.308 | 0.303 | 0.297 | 0.309 | 0.302 | 0.283 |
| Accuracy | 0.462 | 0.457 | 0.478 | 0.481 | 0.453 | 0.453 | 0.471 | 0.472 |
| Robustness | 0.734 | 0.734 | 0.799 | 0.775 | 0.789 | 0.814 | 0.775 | 0.741 |

Table 5: Results on VOT-STB2022 [11].

|  | SwinTrack T-224 | SwinTrack B-384 |
|---|---|---|
| EAO | 0.505 | 0.477 |
| Accuracy | 0.777 | 0.790 |
| Robustness | 0.790 | 0.759 |

## 5 Quantitative Analysis of the Effectiveness of Motion Token

To give a further analysis of the effectiveness of motion token, we provide the success plot (Fig. 4) and precision plot (Fig. 5) on LaSOT test set, and the success AUC score under different attributes of LaSOT test set in Fig. 6. The success plot and the precision plot show that the motion token improves the performance of the trackers by boosting robustness. While the Fig. 6 further points out that the motion token can assist the tracker to recover from a failure state when the vision features are not reliable like an object is getting out of view or fully occluded by other objects.

## 6 Response Visualization for Qualitative Analysis

We provide the heatmap visualization of the response map generated by the IoU-aware classification branch of the head in our SwinTrack-B-384 in Fig. 7. The visualized sequences are from LaSOT$_{ext}$ [5], with challenges include fast motion, full occlusion, hard distractor, *etc*. The results demonstrate the great discriminative power of our tracker. Many trackers will show a multi-peak on the response map when the target object is occluded or multiple similar objects exist. With the vision-motion integrated Transformer architecture, our tracker eases such phenomenon.

## 7 Failure Case

We show some typical failure cases of our tracker (SwinTrack-B-384 on LaSOT$_{ext}$ [5] and VOT-STB2022 [11]) in Fig. 8. The first case suffers from a mixture of low resolution, fast motion, and background clutter. The second case suffers from a fast occlusion by a distractor. The third case suffers from the non-semantic target.

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

Figure 1: Comparison to the state-of-the-art trackers on LaSOT [6] Test set using success (SUC) AUC score.

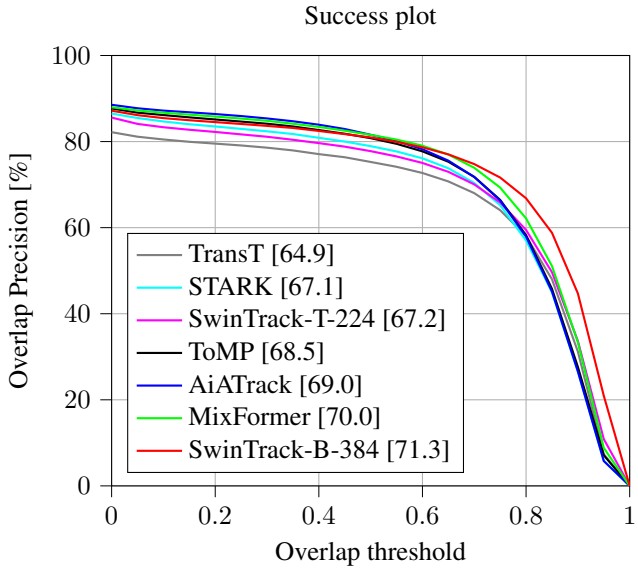

Figure 2: Comparison to the state-of-the-art trackers on LaSOT [6] Test set using precision (PRE) AUC score.

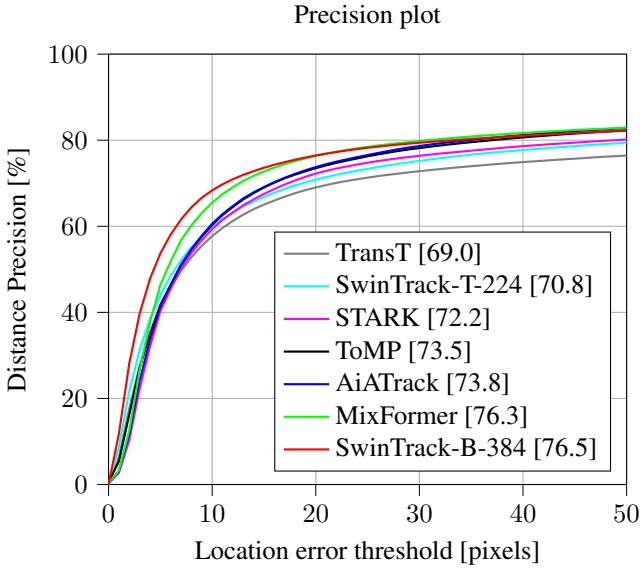

Figure 3: Comparison to the state-of-the-art trackers using success (SUC) AUC score under different attributes of LaSOT [6] Test set.

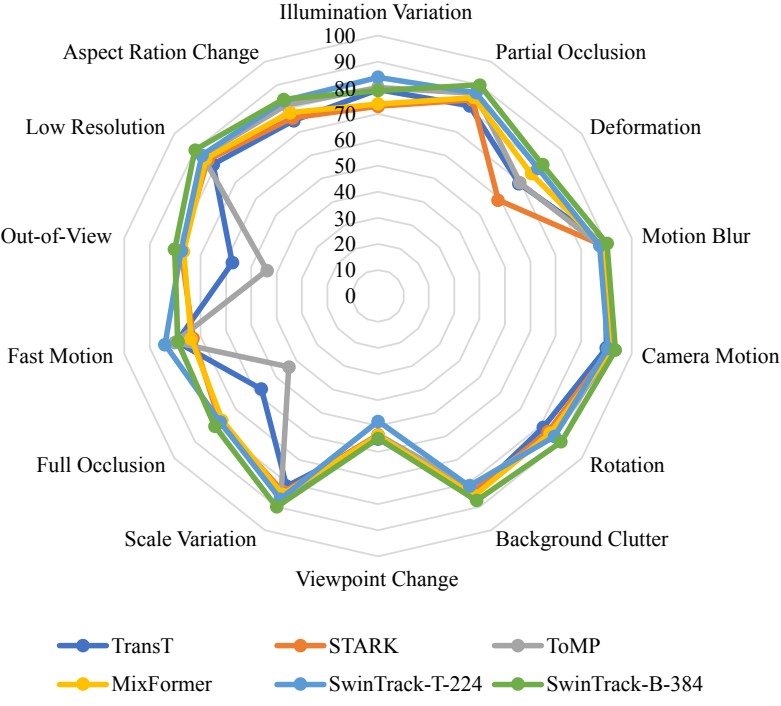

Figure 4: Success (SUC) AUC score on LaSOT [6] Test set assessing the effectiveness of the motion token.

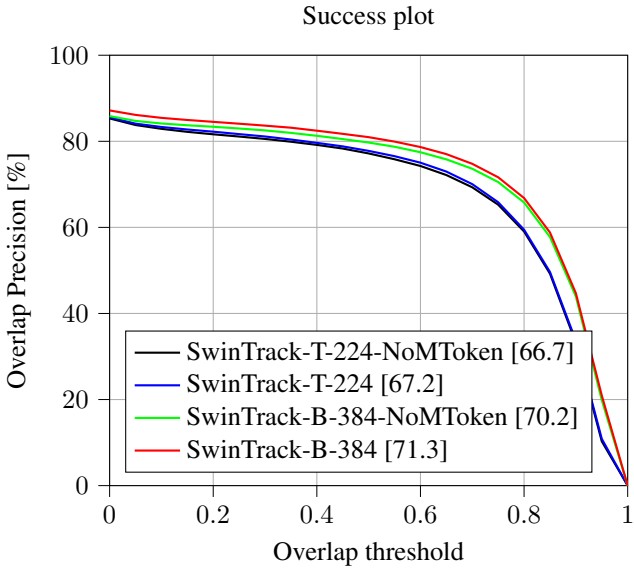

Figure 5: Precision (PRE) AUC score on LaSOT [6] Test set assessing the effectiveness of the motion token.

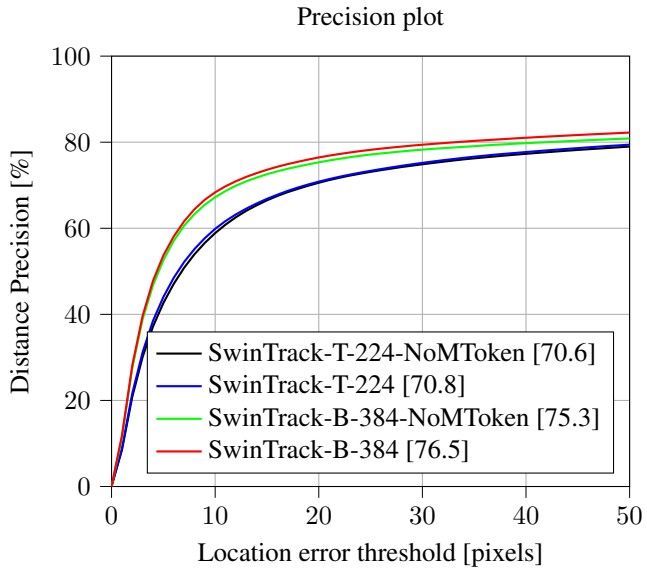

Figure 6: Success (SUC) AUC score under different attributes of LaSOT [6] Test set assessing the effectiveness of the motion token.

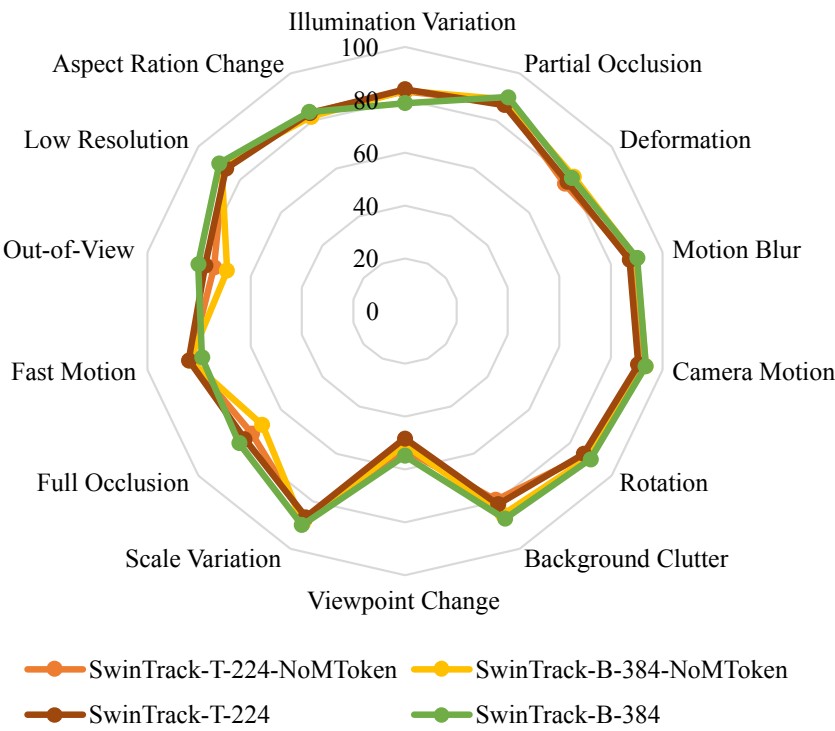

Figure 7: Heatmap visualization of the tracking response map of our SwinTrack-B-384 on LaSOT$_{ext}$ [5]. The odd rows visualize the search region patches with ground-truth bounding box (in red rectangles). The even rows visualize the search region patches blended with the heatmap visualization of the response map. The sequences and challenges involved: atv-10 (POC, ROT, VC, SV, LR, ARC), wingsuit-10 (CM, BC, VC, SV, FOC, LR, ARC), rhino-9 (DEF, SV, ARC) and misc-3 (POC, MB, ROT, BC, SV, FOC, FM, LR).

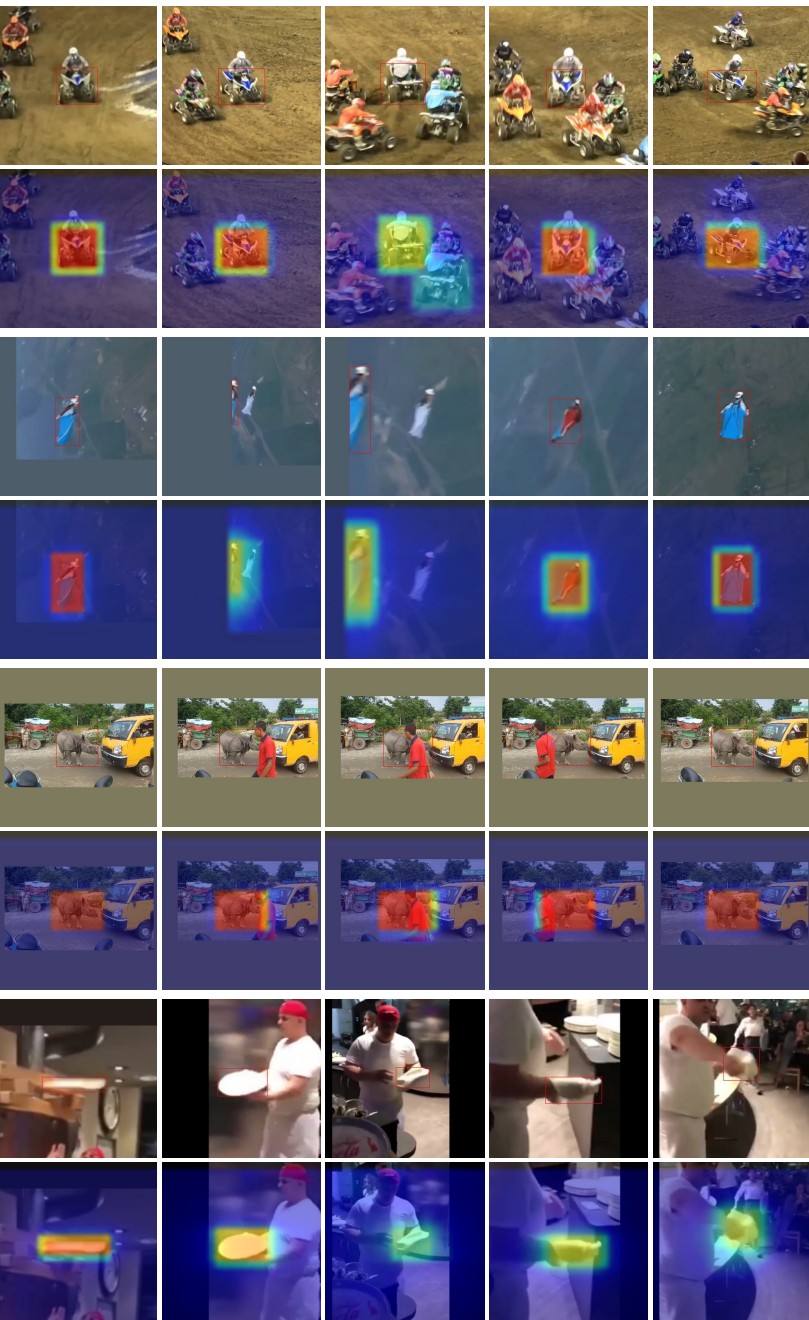

Figure 8: Heatmap visualization of the failure cases. The organizational form is the same as Fig. 7. The sequences and challenges involved: badminton-3 in LaSOT$_{ext}$ (MB, SV, FOC, FM, OV, LR, ARC), skatingshoe-2 in LaSOT$_{ext}$ (POC, MB, ROT, BC, SV, FOC, FM, LR, ARC) and conduction1 (non-semantic target) in VOT-STB2022.[1]

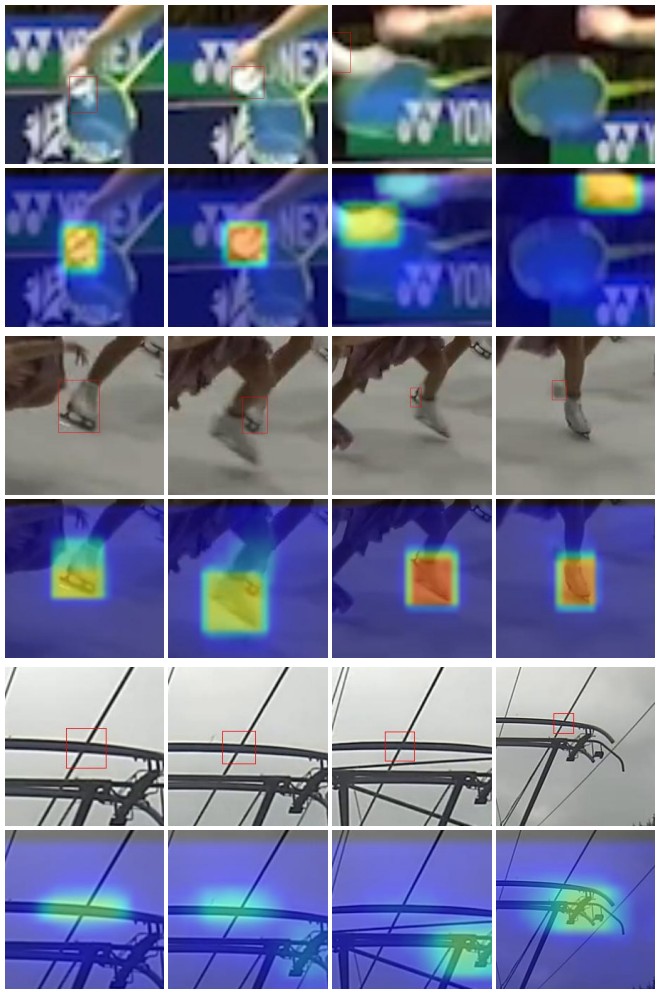

[1]IV: Illumination Variation, POC: Partial Occlusion, DEF: Deformation, MB: Motion Blur, CM: Camera Motion, ROT: Rotation, BC: Background Clutter, VC: Viewpoint Change, SV: Scale Variation, FOC: Full Occlusion, FM: Fast Motion, OV: Out-of-View, LR: Low Resolution, ARC: Aspect Ration Change