# OpenReview forum: "SwinTrack: A Simple and Strong Baseline for Transformer Tracking"
_NeurIPS.cc/2022/Conference — NeurIPS 2022 Accept_

### Official Review · Reviewer_eHVZ · 2022-07-02

**Rating:** 9
**Confidence:** 5
**Soundness:** 3 good
**Presentation:** 3 good
**Contribution:** 3 good

**Summary:**

This paper proposes a fully-attentional tracker where both the representation learning and the feature fusion modules are based on the Transformer. Besides, a motion token is introduced to embed the historical target trajectory, further improving the tracking performance. Extensive experiments and ablation study on several large-scale tracking benchmarks are performed to demonstrate the effectiveness and efficiency of the proposed approach.

**Questions:**

Please explain the aforementioned two weakness issues.

**Ethics Review Area:**

["I don’t know"]

**Strengths And Weaknesses:**

For the strengths, 1) Compared with recent Transformer-based methods that use the Transformer to perform the feature fusion, the proposed tracking framework takes advantage of the Transformer on both the representation and feature fusion modules, which is more thorough and more reasonable.
2) The design of introducing the motion token is novel and effective for tracking.
3) Experiments, especially the ablation study are sufficient, the effectiveness of each designed module is proved.

For the weaknesses, 1) For the motion token, the template-search region pair is based on the single frame, while the object trajectory is sequential. To make the trajectory invariant to the cropping, do all bounding boxes in one trajectory perform the same transformation, i.e., based on the cropping of the current frame? This point is somewhat confusing.
2) The fixed sampling interval of the trajectory is set to 15 and 8 for different datasets, is it the same setting for the training and inference phase? Besides, will it make a significant performance fluctuation when increasing or decreasing this value? I think the setting of this value needs to be explained.

---

> ### Author Response · Authors · 2022-08-02
> **Response To Reviewer eHVZ**
>
> We appreciate the thoughtful and encouraging comments. Below, we provide our responses to address the concerns that were raised.
> ***
> ***Q1: To make the trajectory invariant to the cropping, do all bounding boxes in one trajectory perform the same transformation, i.e., based on the cropping of the current frame? This point is somewhat confusing.***
>
> **A1:** Sorry for the confusion. Your understanding is correct. Normally, in each frame, the Siamese tracker (our tracker also belongs to Siamese tracking) will crop out a search region for tracking. Therefore, the tracker does not have the information beyond the boundary of cropped search region patch in current frame. We apply a coordinate transformation on the historical object trajectory to make it consistent with the coordinate system in the cropped search region image patch. We will clarify this point to make it more clear in the revision. Thanks.
>
> ***
> ***Q2: The fixed sampling interval of the trajectory is set to 15 and 8 for different datasets, is it the same setting for the training and inference phase?***
>
> **A2:** Yes, the sampling interval is fixed to 15 for the training and inference phases in all the experiments except for GOT-10k. In implementation, we assume the frame rate of the video is standard 30 frames per second (fps) by default. Then we perform sampling twice per second, so the sampling interval is 15, in both training and inference stages. For experiments on GOT-10k, since it has a smaller frame rate with 10 fps, we fix the sampling interval to 8, in both training and inference phase. We will further clarify this in revision. Thanks.
>
> ***
>
> ***Q3: Will the sampling interval make a significant performance fluctuation when increasing or decreasing this value? I think the setting of this value needs to be explained.***
>
> **A3:** Thanks, and we totally agree that it is necessary to ablate the sampling interval on the final performance. As suggested, we have conducted experiments on LaSOT using SwinTrack-T-224 with different sampling interval values. The results are reported in Tab. #1. From Tab. #1, we can see that increasing or decreasing the sampling interval does not impact the final performance very much. Based on the experimental results, we set the value of sampling interval to 15. We will include the results with analysis in revision. Again, thanks.
>
> **Table #1:** Ablation study on different values of sampling interval.
>
> |Sampling interval | 60    | 15 | 8|
> |------------------------|-------|----|-----|
> |SUC on LaSOT    | 66.9 | 67.2 | 66.8|
>
> ***

---

### Official Review · Reviewer_iVQV · 2022-07-02

**Rating:** 4
**Confidence:** 4
**Soundness:** 2 fair
**Presentation:** 2 fair
**Contribution:** 2 fair

**Summary:**

The authors propose an interesting Siamese architecture for single object tracking based on the Transformer model. The encoder is divided into the two feature transforms and a transformer encoding step where correlations of spatiotemporal features on the concatenated branches are made. Instead of CNNs the authors suggest SwinTransformers as branches with shared weights. In the encoding part of the Transformer motion information is added to the stream of features by assuming smooth motion of the object and by maintaining a limited history of the BBs of the object. BBs are regressed with the output embeddings. The experiments show state-of-the-art results on large-scale tracking datasets.

**Questions:**

Could you explain in more detail why it is beneficial to neglect learnable queries in the model design?
Why did you not test your method on the VOT challenges? They give experimental results for the different challenges / attributes of tracking.
How would your method work when an object abruptly changes the direction of motion as you assume motion consistency?

**Limitations:**

No.

**Strengths And Weaknesses:**

The idea to use SwinTransformer instead of a CNN in the branches seems to be novel and also seems to be fruitful at least in terms of algorithmic efficiency and surpassing the 70% limit in LASOT (Fig. 1). Concatenating the features of the branches and then applying a Transformer like attentional mechanism to extract embeddings for which BBs can be regressed is then straightforward [5, 36]. The use of non-learnable queries might be a novel change in the design and beneficial in terms of interpreting the tracking problem as classification problem.
Motion consistency restricts the method to track objects properly e.g. when it comes to abrupt motions. Although the common experimental procedure on standard datasets including ablation studies has been conducted, the experiments show little insight into the effects of assumptions and design decisions. While overall benchmark scores and leave-one-out results give an answer to the usefulness of (parts of) the method compared to others, the methodology gives very little insight into the challenges and the potential causes of low performance, e.g. in case of rapid motion, occlusion, illumination changes, etc. I very much miss a deeper qualitative analysis of the model and understanding of its behaviour when it comes to the challenges of tracking.
The paper is well structured, however I recommend to proof-read the document as there are typo errors in the text, e.g. line 120. The English needs to be improved, e.g. line 163: “chopped off” sounds flappy. Technically the paper needs also to be improved, e.g. line 145, for n=0 T_traj is not defined properly, for n=1 T_traj = {o_c-delta, o_c, o_c-delta} which makes no sense.

---

> ### Author Response · Authors · 2022-08-02
> **Response To Reviewer iVQV (part 3)**
>
> ***
> ***Q4: Could you explain in more detail why it is beneficial to neglect learnable queries in the model design?***
>
> **A4:** Sorry for the confusion. The main reason is that it is unclear for Siamese trackers (our tracker belongs to Siamese tracking) to utilize query. Query is applied in Transformer-based object detection because the detector needs to generate the final detection results from a pre-defined number of different input embeddings. To such end, each of the query can focus on a specific kind of object or a particular region in the image, which makes the usage of query reasonable for detection task. In the task of visual tracking, however, we can leverage the encoder of Transformer to fuse the template and search region feature tokens to force the tracker focus on finding the given target template, which naturally forms the standard Siamese tracking architecture and has been proven to be capable of learning discriminative representation as in many previous successful Siamese trackers. The introduction of query may break the design of Siamese architecture and thus degrades the performance, as shown in our experiment (please refer to Table 3 in the main text for the experimental comparison w/o and w/ query and see lines 321-326 for related discussion).
>
> We will include the above explanation in revision. Thanks.
> ***
> ***Q5: Test the tracker on VOT challenges.***
>
> **A5:** As suggested, we have conducted extra experiment on VOT-20 and VOT-22 using SwinTrack-B-384. The results are reported in the setting of bounding box on VOT, as shown in Tab. #4. We will include the results and comparison with other methods in revision. Thanks.
>
> **Table #4:** Experiments on VOT-20 and VOT-22 using the bounding box setting.
>
> |     VOT-20    |     VOT-20    |     VOT-20    |     VOT-22    |     VOT-22    |     VOT-22    |
> |---------------|---------------|---------------|---------------|---------------|---------------|
> |     EAO       |     Acc.      |     Rob.      |     EAO       |     Acc.      |     Rob.      |
> |     0.283     |     0.472     |     0.741     |     0.477     |     0.790     |     0.795     |
> ***
> ***
> ***Q6: How would your method work when an object abruptly changes the direction of motion as you assume motion consistency?***
>
> **A6:** Thanks for this insightful comment. Yes, the object may abruptly change the direction of motion. In this situation, the assumption of motion consistency may not be accurate. However, the proposed tracker can still locate the target object by leveraging the complementary visual appearance information of the target. Please note that, the motion token functions as a supplementary cue to the vision features for improving tracking. If the motion is not consistent due to abrupt target motion, we can still use the important visual information for localization. Besides, in most time, the target moves smoothly, and its motion is consistent, which benefits the tracker from our motion token for improvement.
> We thank the reviewer again and will clarify this point in revision.
> ***

---

> ### Author Response · Authors · 2022-08-02
> **Response To Reviewer iVQV (part 2)**
>
> ***
> ***Q2: The methodology gives very little insight into the challenges and the potential causes of low performance, e.g., in case of rapid motion, occlusion, illumination changes, etc. I very much miss a deeper qualitative analysis of the model and understanding of its behavior when it comes to the challenges of tracking.***
>
> **A2:** We appreciate this constructive comment. As suggested, in order to better understand the proposed method, we qualitatively and quantitatively show and compare our tracker with other state-of-the-arts on different challenges (or the so-called attributes) such as occlusion, rapid motion, etc. Due to the format restriction, we cannot show the qualitative results in figures here, please kindly refer to Fig.7 in Sec. F in the the updated supplementary material for the qualitative results on different attributes. Here, we demonstrate the quantitative results of our method on different challenges and its comparison with other sota trackers, as in Tab. #1. From Tab. #1, we can see that our SwinTrack-B-384 performs more robustly on most challenges (11 out of 14) such as partial or full occlusion, motion blur, scale variation, low-resolution, etc.
>
> We thank the reviewer again for this helpful comment and will include both qualitative analysis (please refer to the Fig. 7 in Sec. F in the updated supplementary material) and quantitative comparison in revision.
>
> **Table #1:** Quantitative results of SwinTrack and comparisons to other state-of-the-art methods on different challenges on LaSOT. IV: Illumination Variation; POC: Partial Occlusion; DEF: Deformation; MB: Motion Blur; CM: Camera Motion; ROT: Rotation; BC: Background Clutter; VC: Viewpoint Change; SV: Scale Variation; FOC: Full Occlusion; OV: Out-of-View; LR: Low Resolution; ARC: Aspect Ratio Change.
>
> |     Challenges    |     TransT    |     STARK    |     ToMP    |     MixFormer    |     SwinTrack-T-224    |     SwinTrack-B-384    |
> |-------------------|---------------|--------------|-------------|------------------|------------------------|------------------------|
> |     IV            |     79.3      |     72.9     |     80.1    |     73.8         |     84.0               |     78.9               |
> |     POC           |     81.1      |     83.6     |     86.3    |     84.6         |     86.6               |     89.9               |
> |     DEF           |     69.2      |     59.0     |     69.7    |     75.3         |     78.6               |     80.9               |
> |     MB            |     89.4      |     87.9     |     87.5    |     88.2         |     87.3               |     90.2               |
> |     CM            |     90.0      |     91.4     |     92.3    |     92.4         |     90.6               |     93.5               |
> |     ROT           |     81.1      |     83.6     |     86.3    |     84.6         |     86.6               |     89.9               |
> |     BC            |     84.6      |     83.8     |     80.9    |     85.3         |     81.1               |     87.2               |
> |     VC            |     54.9      |     53.3     |     53.6    |     53.6         |     48.2               |     54.8               |
> |     SV            |     81.1      |     83.6     |     86.3    |     84.6         |     86.6               |     89.9               |
> |     FOC           |     57.4      |     77.7     |     43.8    |     76.7         |     77.7               |     80.3               |
> |     FM            |     79.3      |     72.9     |     80.1    |     73.8         |     84.0               |     78.9               |
> |     OV            |     57.4      |     77.7     |     43.8    |     76.7         |     77.7               |     80.3               |
> |     LR            |     81.1      |     83.6     |     86.3    |     84.6         |     86.6               |     89.9               |
> |     ARC           |     74.7      |     75.9     |     81.4    |     78.2         |     83.3               |     83.8               |
> ***
> ***
> ***Q3: Proof-read the document as there are typo errors.***
>
> **A3:** Thanks. We will carefully polish the manuscript and fix all the typos in revision.
> ***

---

> ### Author Response · Authors · 2022-08-02
> **Response To Reviewer iVQV (part 1)**
>
> Thank you for providing valuable and thoughtful comments on our work. Below, we address the concerns and remain committed to clarifying further questions that may arise during the discussion period.
>
> ***
> ***Q1: Although the common experimental procedure on standard datasets including ablation studies has been conducted, the experiments show little insight into the effects of assumptions and design decisions.***
>
> **A1:** The reviewer has raised an important point. However, we believe that our extensive experiments have verified the effectiveness of many assumptions and design choices in this work. In Sec. 4.3, we have conducted various validation experiments. For example, in Table 3 of the main text, we have studies and analyzed two different feature fusion strategies (line 314-320), and empirically show that the concatenation-based fusion is a better choice, consistent with our assumption in Sec. 3.3 (line 176-186). In addition, we also discuss different decoders, position encoding methods and loss function with analysis. This analysis shows the insight of our design decisions. Besides, in Table 4, we analyze in-depth the effectiveness of the proposed motion token. We believe that all this analysis and discussion can reflect the insights of our approach in achieving a simple, yet highly powerful tracking framework, as mentioned by **Reviewer 6BWX** in the strengths (here is the quote from **Reviewer 6BWX** “… *On the contrary, I believe that this paper brings substantial value to the tracking field by consolidating existing techniques, while investigating important details in order to achieve a simple, yet highly powerful tracking framework. Importantly the authors provide valuable insights when motivating their approach and comparing to other techniques (modifications of fusion, transformer architecture, losses, etc.* …)”).
> ***

---

### Official Review · Reviewer_6BWX · 2022-07-07

**Rating:** 6
**Confidence:** 5
**Soundness:** 3 good
**Presentation:** 3 good
**Contribution:** 3 good

**Summary:**

The paper addresses the problem of general object tracking, where the target object is given during test time. The authors propose a simple but strong baseline approach based on a Swin Transformer image encoder and a standard transformer decoder. Further, the paper introduces a motion token, which encodes past bounding boxes in order to improve the final prediction. The authors also revise the losses employed to train the tracker. Very strong results are reported on 5 popular benchmarks.

**Questions:**

See weaknesses.

**Limitations:**

I could not find discussion of negative social impact or limitations. It would be good to add these.

**Strengths And Weaknesses:**

Before dwelling into the strengths and weaknesses, I want to discuss the novelty aspect of this paper. On one hand, the novelty is not remarkable: the authors are adopting the Swin backbone, while concatenation fusion is also used in Stark. However, the novelty is not limited to these aspects. On the contrary, I believe that this paper brings substantial value to the tracking field by consolidating existing techniques, while investigating important details in order to achieve a simple, yet highly powerful tracking framework. Importantly the authors provide valuable insights when motivating their approach and comparing to other techniques (modifications of fusion, transformer architecture, losses, etc.). Lastly, I find the motion token a very interesting novelty that could reopen a long-forgotten direction in tracking, namely exploiting motion prediction and other dynamic information. Although seemingly incremental at the first glance, I consider the novelty to be significant based on how much this paper advances the useful knowledge in the field.

Other strong points of the paper are:

• Very strong results, clearly SOTA.

•	Simple and elegant architecture.

•	Insightful discussions.

•	Method and motivation are easy to follow.

•	Interesting ablative experiments on multiple datasets.

•	Relatively fast frame-rates.


Weaknesses:

1. Details regarding pre-training are missing. I assume that the authors use ImageNet-22k pre-training, while most trackers use ImageNet-1k. This has shown to give about 2-3% on LaSOT in previous papers. The authors should therefore analyze this in a separate experiment.

2. By looking into the more detailed results in the supplementary material, it seems that the improvements mostly stem from increased accuracy. That is, better bounding box regression. The robustness (low overlap scores in the success plot) seems to be on par with recent trackers. While accuracy is also important, the major challenge in tracking is to improve the robustness. It is important to discuss these aspects in the paper in order to understand where and how SwinTrack performs better compared to other trackers. Moreover, please add more high-performing trackers to the success plots in the supplementary material.

3. Unfortunately, the design of the motion token is motivated. For instance, why are the past box encodings concatenated in the channel dimension, and not processed in some other manner? Why add it as a single token in the transformer, instead of one per box?

4. Results on UAV and VOT should be added, even if the tracker does not beat SOTA.

5. There are quite a few language mistakes.

---

> ### Author Response · Authors · 2022-08-02
> **Response To Reviewer 6BWX (part 4)**
>
> ***
> ***Q6: Add results on UAV and VOT, even if the tracker does not beat SOTA.***
>
> **A6:** As suggested, we have conducted extra experiments on UAV123 and VOT-20 and VOT-22 using SwinTrack-B-384. The results are reported in Tab. #4. Please note that, on VOT-20 and VOT-22, we use the setting of bounding box for evaluation. We will include the results and comparison with other methods in revision. Thanks.
>
> ***
>
> **Table #4:** Experiments on UAV123, VOT-20 and VOT-22. Note that, on VOT-20 and VOT-22, we use the setting of bounding box for evaluation.
>
> |     UAV123     |     UAV123    |     VOT-20    |     VOT-20    |     VOT-20    |     VOT-22    |     VOT-22    |     VOT-22    |
> |----------------|---------------|---------------|---------------|---------------|---------------|---------------|---------------|
> |     SUC (%)    |     P (%)     |     EAO       |     Acc.      |     Rob.       |     EAO       |     Acc.      |     Rob.      |
> |     70.5       |     90.0      |     0.283     |     0.472     |     0.741     |     0.477     |     0.790     |     0.795     |
>
> ***
>
> ***Q7: Language mistakes.***
>
> **A7:** Thanks for pointing this issue out. We will carefully polish the manuscript and fix these mistakes in revision.
>
> ***

---

> ### Author Response · Authors · 2022-08-02
> **Response To Reviewer 6BWX (part 3)**
>
> ***
> ***Q3: Add more high-performing trackers to the success plots in the supplementary material.***
>
> **A3:** Thanks for this comment. As suggested, we add six very recent high-performing trackers to the success plots in the supplementary material, including OSTrack-384 [*1] (ECCV 2022), Unicorn [*2] (ECCV 2022), AiATrack [*3] (ECCV 2022), MixFormer [*4] (CVPR 2022), ToMP [*5] (CVPR 2022) and SBT [*6] (CVPR 2022). Due to the format restriction, we show the new success plots in the following Tab. #3, where our SwinTrack-B-384 achieves the best performance. We will update the success plots in the supplementary material in revision. Again, thanks.
>
> ***
>
> **Table #3:** Adding more high-performing trackers in the success plots in the supplementary material. The rows in bold fonts represent the newly added high-performing trackers (all in 2022). Note that, here we report the best results for each tracker quoted from their official papers.
>
> |     Tracker            |     Success   score (%)    |
> |------------------------|----------------------------|
> |     DiMP               |     56.9                   |
> |     STMTrack           |     60.6                   |
> |     TrDiMP             |     63.9                   |
> |     TransT             |     64.9                   |
> |     STARK              |     67.1                   |
> |     **SBT**                |     **66.7**                   |
> |     **ToMP**               |     **68.5**                   |
> |     **MixFormer**          |     **70.1**                   |
> |     **AiATrack**           |     **69.0**                   |
> |     **Unicorn**            |     **68.5**                   |
> |     **OSTrack-384**        |     **71.1**                   |
> |     SwinTrack-T-224    |     67.2                  |
> |     SwinTrack-B-384    |     71.3                   |
>
> [*1] Ye et al., Joint Feature Learning and Relation Modeling for Tracking: A One-Stream Framework, ECCV, 2022.
>
> [*2] Yan et al., Unicorn: Towards Grand Unification of Object Tracking, ECCV, 2022.
>
> [*3] Gao et al., UAiATrack: Attention in Attention for Transformer Visual Tracking, ECCV, 2022.
>
> [*4] Cui et al., MixFormer: End-to-End Tracking with Iterative Mixed Attention, CVPR, 2022.
>
> [*5] Mayer et al., Transforming Model Prediction for Tracking, CVPR, 2022.
>
> [*6] Xie et al., Correlation-Aware Deep Tracking, CVPR, 2022.
>
> ***
>
> ***Q4: Why are the past box encodings concatenated in the channel dimension, and not processed in some other manner?***
>
> **A4:** Thanks for this comment. The concatenation strategy is determined based on our empirical results. Concatenating the past box encodings leads to the best performance. We also tried other strategies, such as summing up the past box embeddings to generate the motion token, and the results are degenerated. We argue that summing up manner may result in over-parameterization on the coordinate embeddings, causing performance drop. We will clarify this point in revision. Again, thanks.
>
> ***
>
> ***Q5: Why add the motion token as a single token in the transformer, instead of one per box?***
>
> **A5:** Thanks for this insightful comment. The reasons are two-fold. First, motion token is designed to learn the temporal motion representation by considering the relationship among all historical box embeddings. If each box is treated as a token and added to the decoder along with visual features, it will be ineffective in learning the temporal motion representation. To verify this, we have conducted an extra experiment using SwinTrack-T-224 by treating each box as a token and feed all these tokens into the decoder for tracking. When using each box as a token and feeding them into decoder, the SUC score on LaSOT is 64.5%, which is much lower than the SUC score of 67.2% when learning a single motion token. In addition, another reason is to avoid high computation in the decoder. If each box is treated as a token in the decoder, the computation will be much heavier then feeding a single temporal motion token. Considering these two reasons, we design to learn a single motion token for tracking. We will add more explanation to clarify this in revision. Again, thanks.
> ***

---

> ### Author Response · Authors · 2022-08-02
> **Response To Reviewer 6BWX (part 2)**
>
> ***
> ***Q2: Discuss where and how SwinTrack performs better compared to other trackers for better understanding.***
>
> **A2:** Thanks for this insightful comment and we totally agree with the reviewer. As suggested, in addition to the overall performance, we further analyze the results of our SwinTrack and its comparison to other state-of-the-art trackers on each challenge. We show the results in Tab. #2, where SwinTrack demonstrates much better performance on specific challenges. On most challenges (11 out of 14), SwinTrack-B-384 shows the best results. For example, for partial and full occlusion, it shows better robustness owing to the proposed motion token that re-captures target after recovery of object. We will include the results here with analysis in revision. Again, thanks.
>
> **Table #2:** Quantitative results of SwinTrack and comparisons to other state-of-the-art methods on different challenges on LaSOT. IV: Illumination Variation; POC: Partial Occlusion; DEF: Deformation; MB: Motion Blur; CM: Camera Motion; ROT: Rotation; BC: Background Clutter; VC: Viewpoint Change; SV: Scale Variation; FOC: Full Occlusion; OV: Out-of-View; LR: Low Resolution; ARC: Aspect Ratio Change.
>
> |     Challenges    |     TransT    |     STARK    |     ToMP    |     MixFormer    |     SwinTrack-T-224    |     SwinTrack-B-384    |
> |-------------------|---------------|--------------|-------------|------------------|------------------------|------------------------|
> |     IV            |     79.3      |     72.9     |     80.1    |     73.8         |     84.0               |     78.9               |
> |     POC           |     81.1      |     83.6     |     86.3    |     84.6         |     86.6               |     89.9               |
> |     DEF           |     69.2      |     59.0     |     69.7    |     75.3         |     78.6               |     80.9               |
> |     MB            |     89.4      |     87.9     |     87.5    |     88.2         |     87.3               |     90.2               |
> |     CM            |     90.0      |     91.4     |     92.3    |     92.4         |     90.6               |     93.5               |
> |     ROT           |     81.1      |     83.6     |     86.3    |     84.6         |     86.6               |     89.9               |
> |     BC            |     84.6      |     83.8     |     80.9    |     85.3         |     81.1               |     87.2               |
> |     VC            |     54.9      |     53.3     |     53.6    |     53.6         |     48.2               |     54.8               |
> |     SV            |     81.1      |     83.6     |     86.3    |     84.6         |     86.6               |     89.9               |
> |     FOC           |     57.4      |     77.7     |     43.8    |     76.7         |     77.7               |     80.3               |
> |     FM            |     79.3      |     72.9     |     80.1    |     73.8         |     84.0               |     78.9               |
> |     OV            |     57.4      |     77.7     |     43.8    |     76.7         |     77.7               |     80.3               |
> |     LR            |     81.1      |     83.6     |     86.3    |     84.6         |     86.6               |     89.9               |
> |     ARC           |     74.7      |     75.9     |     81.4    |     78.2         |     83.3               |     83.8               |
> ***

---

> ### Author Response · Authors · 2022-08-02
> **Response To Reviewer 6BWX (part 1)**
>
> We appreciate your careful review and thoughtful and encouraging comments on our work. Below, we address the concerns that were raised.
> ***
> ***Q1: (1) Details regarding pre-training are missing. (2) Analyze SwinTrack with different ImageNet-22k and ImageNet-1k pre-training methods in a separate experiment.***
>
> **A1:** **(1)** Sorry about this and we will add the details in revision. Regarding the pre-training, we simply borrow the pre-trained models on ImageNet provided by Swin Transformer. In specific, the backbone of our SwinTrack-T-224 is pre-trained on ImageNet-1k and the backbone of SwinTrack-B-384 is pre-trained on ImageNet-22k.
>
> **(2)** We appreciate this constructive comment. As suggested, we have conducted experiments, following the settings in the ablation study in the submission (motion token is not used and the result on GOT-10k is trained on the full dataset), on SwinTrack-T-224 and SwinTrack-B-384 with ImageNet-22k and ImageNet-1k pre-trainings, respectively. The results are listed in Tab. #1. From Tab. #1, we can observe that, for smaller model SwinTrack-T-224 (23M # parameters), pre-training on ImageNet-22k brings small improvements on LaSOT (+0.6%) and TrackingNet (+0.4%) but degrades the performance on GOT-10k (-1.4%). For larger model SwinTrack-B-384 (91M # parameters), pre-training on ImageNet-22k shows significant performance gains on LaSOT (+2.2%) and GOT-10k (+3.0%) but slightly degrades the result on TrackingNet (-0.6%). On LaSOT_ext, ImageNet-22k shows a performance degradation on smaller model SwinTrack-T-224 (-0.9%) and brings small improvements on larger model SwinTrack-B-384 (+0.2%).
>
> We thank the reviewer again for this helpful comment and will include all the results with analysis in revision.
>
> **Table #1:** Experiments on SwinTrack-T-224 and SwinTrack-B-384 with ImageNet-22k and ImageNet-1k pre-trainings, respectively.
>
> |        |         | LaSOT        | LaSOT_ext | TrackingNet | GOT-10k |
> |-----------------|--------------|-----------|-------------|---------|---------|
> |                 | Pre-training | SUC (%)   | SUC (%)     | SUC (%) | mAO (%) |
> | SwinTrack-T-224 | ImageNet-1k  | 66.7      | 46.9        | 80.8    | 70.9    |
> | SwinTrack-T-224 | ImageNet-22k | 67.3      | 46.0        | 81.2    | 69.5    |
> | SwinTrack-B-384 | ImageNet-1k  | 68.0      | 47.3        | 83.8    | 71.8    |
> | SwinTrack-B-384 | ImageNet-22k | 70.2      | 47.5        | 83.2    | 74.8    |
>
> ***

---

### Official Review · Reviewer_ZRLJ · 2022-07-11

**Rating:** 7
**Confidence:** 5
**Soundness:** 4 excellent
**Presentation:** 3 good
**Contribution:** 3 good

**Summary:**

This paper presents a simple yet strong baseline tracker, named SwinTrack, for single object tracking (SOT). SwinTrack adopts the classic siamese structure, and uses Swin Transformer for both feature extraction and fusion of template and search region. A special motion token which encodes the past trajectory is introduced to further boost the tracking performance. SwinTrack is evaluated on major SOT benchmarks and achieves SOTA performance.

**Questions:**

One main contribution of SwinTrack is to use Transformer network not only for feature fusion but for feature extraction. This is quite obvious since Transformer-based vision feature has shown its advantages over CNN-based feature in numerous scenarios and downstream tasks. One question I have been wondering is  why previous Transformer-based trackers adopt CNN-Transformer hybrid architecture? Are there any obstacles in using Transformer for feature extraction? Is there any secret weapon that the authors use to make it work?

**Limitations:**

This reviewer does not see potential negative societal impact of this work.

**Strengths And Weaknesses:**

Single object tracking (SOT) is an important problem in computer vision and for video understanding in particular. The proposed method is technically sound and achieves SOTA performance on major SOT benchmarks. Although most components in SwinTrack are existing ones and cannot be said novel, I appreciate the system work which puts them together in a proper way and eventually achieves good results. In particular, the design choices to use concatenation-based fusion over cross-attention-based fusion and not to use query-based decoder are reasonable and well-justified by ablation studies.
The motion token design is new, based on my knowledge. According to the ablation, it achieves non-negligible gain, making it a good added component to the tracker.

---

> ### Author Response · Authors · 2022-08-02
> **Response To Reviewer ZRLJ**
>
> Thank you for the encouraging comments and constructive suggestions. Below, we provide responses to address the questions and concerns.
>
> ***
> ***Q1: Why do previous Transformer-based trackers adopt CNN-Transformer hybrid architecture?***
>
> **A1:** Thanks for this insightful comment. We argue that there are two possible reasons attributing to this. **First**, the focus of most previous trackers (e.g., [5, 32, 36]) is more on improving feature fusion of the target template(s) and the search region in the matching process, instead of on the basic feature learning process. Therefore, they usually use a well-designed CNN backbone for feature extraction and Transformer for feature fusion in the template-search region matching process. **Second**, the vanilla vision Transformer at that moment does not exhibit superior performance and advantages compared to popular CNN backbones such as ResNets (although later it does show better results). Because of the above reasons, previous trackers usually adopt a hybrid CNN-Transformer architecture. We will include the discussion here into the final revision. Again, thanks.
>
> ***
>
> ***Q2: Are there any obstacles in using Transformer for feature extraction?***
>
> **A2:** Thanks for this comment. In our opinion, there are two main obstacles in applying Transformer for feature extraction in tracking. **First**, the feature extraction is highly computationally expensive. The core of Transformer is attention, and its complexity is quadratic, which results in extensive computation in feature extraction in the past. **Second**, Transformer usually requires a lot more data during the training stage and takes longer time to converge in comparison with convolutional networks. Because of this, using convolutional network for feature extraction may be more resource-friendly in tracking.
>
> Fortunately, recent research on Transformer has gradually removed these obstacles for feature extraction, which makes the usage of Transformer easier and significantly improves the performance in various tasks. We believe that our proposed SwinTrack serves as a good example.
>
> We thank the reviewer again for this comment and will incorporate the above analysis in revision.
>
> ***
>
> ***Q3: Is there any secret weapon that the authors use to make it work?***
>
> **A3:** We appreciate this interesting question. The secret weapon is to conduct numerous experiments with extensive analysis to find out each component with the most reasonable structure for visual tracking. For example, we study and analyze different feature fusion strategies (e.g., concatenation-based fusion and cross attention-based fusion) and different decoders, explore and adapt position encoding methods to our task, introduce the novel motion token and so on. All these studies and analysis help us in building the proposed simple yet powerful tracking framework.
> ***

---

### Meta-Review · Area_Chair_WdkN · 2022-08-21

**Recommendation:** Accept
**Confidence:** Certain

**Metareview:**

Authors present a method for single object tracking (SOT) that is entirely comprised of transformers. The architecture is simple:

1) Swin is used to generate embeddings for both template and search region
2) Embeddings are concatenated
3) An encoder transformer performs MHSA of the embeddings.
4) A decoder performs cross-attention from search tokens to template tokens and a special "motion token" which is constructed from a linear operation over the prior motion trajectory relative to the frame.
5) Output token is fed to final layers that perform IoU aware classification and bounding box regression.

Evaluations are performed on 5 SOT datasets, achieving SOTA on all of them.


Pros:
- [AC/R] Important problem, technically sound, and new SOTA on this task.
- [AC/R] New motion token approach is novel and provides significant improvement.
- [AC/R] Simple and elegant architecture
- [AC/R] Insightful discussions
- [AC/R] Clearly written and easy to follow
- [AC/R] Interesting ablations
- [AC/R] High frame rate

Cons:
- [R] Low novelty of transformer approach, but this is negated by the novelty of the motion token.
- [R] Details regarding pretraining are missing. Authors provide in response.
- [R] Motivation of motion token design is not clear. Authors provided further ablations of different implementations of the motion token, showing the current form performs best.
- [R] Provide additional details regarding where and how SwinTrack outperforms other approaches on the benchmarks. Authors provided additional granularity of performance stratifications within the LaSOT benchmark.
- [R] Add more recent high performing trackers. Authors added several methods published in 2022.
- [R] Some additional questions about various details were posed by reviewers, which will all answered by the authors.

The single reviewer with reject recommendation changed to accept in their comments after the discussion period but did not update their score. Given unanimous agreement on accept, AC recommendation is accept.

AC Rating: Strong Accept

**Award:**

Yes

---

### Decision · Program_Chairs · 2022-09-14

Accept